# WHY AND WHEN DEEP IS BETTER THAN SHALLOW: AN IMPLEMENTATION-AGNOSTIC STATE-TRANSITION VIEW OF DEPTH SUPREMACY

## ABSTRACT

Why and when is deep better than shallow? We answer this question in a framework that is agnostic to network implementation. We formulate a deep model as an abstract state-transition semigroup acting on a general metric space, and separate the implementation (e.g., ReLU nets, transformers, and chain-of-thought) from the abstract state transition. We prove a bias-variance decomposition in which the variance depends only on the abstract depth-$k$ network and not on the implementation (Theorem 1). We further split the bounds into output and hidden parts to tie the depth dependence of the variance to the metric entropy of the state-transition semigroup (Theorem 2). We then investigate implementation-free conditions under which the variance grow polynomially or logarithmically with depth (Section 4). Combining these with exponential or polynomial bias decay identifies four canonical bias-variance trade-off regimes (EL/EP/PL/PP) and produces explicit optimal depths $k^*$. Across regimes, $k^* > 1$ typically holds, giving a rigorous form of depth supremacy. The lowest generalization error bound is achieved under the EL regime (exp-decay bias + log-growth variance), explaining why and when deep is better, especially for iterative or hierarchical concept classes such as neural ODEs, diffusion/score-matching models, and chain-of-thought reasoning.

## 1 INTRODUCTION

Deep learning has achieved remarkable empirical success, yet its theoretical foundations remain incomplete. Empirically, deeper architectures excel across modalities, yet classical complexity measures predict that increasing depth should worsen generalization. Prior studies using VC-dimension bounds (Bartlett et al., 2019), norm-based complexity measures (Neyshabur et al., 2015; Bartlett et al., 2017; Golowich et al., 2020), compression arguments (Arora et al., 2018; Suzuki et al., 2020; Lotfi et al., 2022), and nonparametric regression theory (Schmidt-Hieber, 2020; Suzuki, 2019; Nakada & Imaizumi, 2020; Imaizumi & Fukumizu, 2022), have derived estimation error bounds that grows exponentially, polynomially, or logarithmically with depth, but almost always within restrictive architectural assumptions (e.g., ReLU networks under strong norm controls). This leaves open a central question: why and when is deep better than shallow, in a way that is not tied to a particular implementation. We address this gap by recasting deep learning as abstract state-transition models, and by analyzing depth through the geometry and algebra of the transition semigroups.

We start from a bias-variance decomposition in which the variance term is independent of implementation: it depends only on the ideal depth-$k$ hypothesis class $\mathcal{H}_k = H \circ B(k, F)$, where $B(k, F)$ is the word ball of length $k$ in the generator set $F$ (Theorem 1). We then split the Rademacher bound into output and hidden contributions and investigate the hidden-layer effect via the entropy integral (Theorem 2). This reduction shows that the growth of covering numbers of word balls—a property of the state-transition semigroup—governs the depth dependence of variance. We identify general conditions that suppress growth to polynomial or logarithmic order, giving a structural explanation of when deeper models remain statistically benign (Section 4).

Why and when is deep better? We pair these variance results with bias decay that is either exponential or polynomial in depth, yielding four canonical trade-off regimes (EL/EP/PL/PP). We compute the optimal depth $k^*$ in each regime and discuss representative examples (Sections 5-6). Two mes-

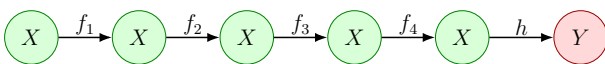

Figure 1: Example of a neural network (depth $k = 4$) in consideration. The input layer is formulated as state space $\mathcal{X}$, the hidden layers as state transition functions $f_i : \mathcal{X} \to \mathcal{X}$, and the output layer as readout function $h : \mathcal{X} \to \mathbb{R}$. The entire network is formulated as a state transition model.

sages follow. First, depth supremacy: across regimes the optimal depth typically satisfies $k^* > 1$, so shallow ($k = 1$) is not optimal in general. Second, the EL regime (exponential bias decay with only logarithmic variance growth) is where depth helps the most, a pattern naturally realized in hierarchical or iterative concept classes—neural ODEs, diffusion/score models, chain-of-thought reasoning—where composition is intrinsic to the data-generation or inference process.

## 2  SETTING

The most important feature of deep neural networks is function composition. Especially in modern AI, there are cases where operations include repeatedly calling an AI model itself, such as autoregression, in-context learning, test-time computation, and diffusion models. Although such operations may not have explicit network implementations, they can also be regarded as a kind of deep structure in the sense of sequential information processing. From this perspective, we formulate neural networks as *state transition models*. Precisely, to cover as wide a range of deep learning models as possible, we formulate the data domain, denoted $\mathcal{X}$, as an arbitrary metric space, and the deep network as a sate-transition model by identifying the data domain $\mathcal{X}$ with state space, hidden layer $f : \mathcal{X} \to \mathcal{X}$ with transition, and output layer $h : \mathcal{X} \to \mathbb{R}$ with observation/readout. (see Fig. 1)

### 2.1  COMMON DATA DOMAINS AND FUNCTION SPACES

We employ *one-based numbering* in this study, such as $\mathbb{N} = \{1, 2, \cdots\}$ and $[n] := \{1, 2, \cdots, n\}$.

Let $(\mathcal{X}, d)$ be a metric space serving as the state space, and let $\mathcal{Y} = \mathbb{R}$ be the observation space. Let $(\mathcal{X} \times \mathcal{Y}, P)$ be a probability space, and let $S_n := \{(X_i, Y_i)\}_{i=1}^n \sim P$ denote an i.i.d. sample.

Let $C(\mathcal{X})$ be the normed space of all continuous real-valued functions, $h : \mathcal{X} \to \mathbb{R}$, equipped with uniform norm $\|h\|_\infty := \sup_{x \in \mathcal{X}} |h(x)|$. Let $C(\mathcal{X}, \mathcal{X})$ be the metric space of all continuous self-maps, $f : \mathcal{X} \to \mathcal{X}$, equipped with uniform metric $d_\infty(f, g) := \sup_{x \in \mathcal{X}} d(f(x), g(x))$. In particular, let $\mathrm{id} : \mathcal{X} \to \mathcal{X}$ denote the identity map. We note that both $C(\mathcal{X})$ and $C(\mathcal{X}, \mathcal{X})$ need not be complete in this study. We summarized (pre)compactness of function sets in Appendix I.

### 2.2  NEURAL NETWORKS (HYPOTHESIS CLASS)

We denote by $F \subset C(\mathcal{X}, \mathcal{X})$ the set of functions corresponding to hidden layers, and by $H \subset C(\mathcal{X})$ the set of functions corresponding to output layers. For example, in the case of a ReLU network, we may take $\mathcal{X}$ to be the Euclidean space $\mathbb{R}^m$ or the cube $[0, 1]^m$, $F$ to be the set of affine transformations with ReLU activations $\{f(x) = \mathrm{ReLU}(Wx) \mid W \in \mathbb{R}^{m \times m}\}$, and $H$ to be the set of linear functions $\{h(x) = w \cdot x \mid w \in \mathbb{R}^m\}$. Since our setting is quite general, we can also take $\mathcal{X}$ to be a graph or a manifold, and $F$ to be CNNs or LLMs.

We then define the depth-$k$ neural network (hypothesis class) as

$$\mathcal{H}_k := H \circ B(k, F). \tag{1}$$

Here, $B(k, F)$ denotes the *word ball*, i.e. the set of all continuous self-maps on $\mathcal{X}$ obtained by composing elements of $F$ at most $k$ times. For example, $B(2, F) = \{\mathrm{id}, f_1, f_2 \circ f_1 : f_1, f_2 \in F\}$, and thus $\mathcal{H}_2 = \{h, h \circ f_1, h \circ f_2 \circ f_1 : h \in H, f_1, f_2 \in F\}$. By construction, $\mathcal{H}_0 = H$, $\mathcal{H}_k \subset \mathcal{H}_{k+1}$, and $\mathcal{H}_k \subset C(\mathcal{X}, \mathcal{X})$. We note when the generator $F$ contains the identity element $\mathrm{id}$, then $F^k = B(k, F)$. So, the following identity always holds: $(F \cup \{\mathrm{id}\})^k = B(k, F)$.

Besides the abstract network model $\mathcal{H}_k$, we introduce another class $\mathcal{H}_{\mathrm{imp}} \subset C(\mathcal{X}, \mathcal{X})$ called the *implementation model*. While $\mathcal{H}_k = H \circ B(k, F)$ is not supposed to have implementation, $\mathcal{H}_{\mathrm{imp}}$ is supposed to have specific implementations such as ReLU networks, ConvNets, and LLMs.

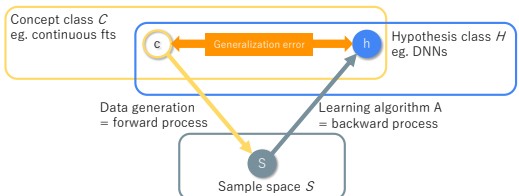

Figure 2: Framework of machine learning in consideration. The *concept class* $\mathcal{C}$ is the class of (unseen) data generators called concept, the *hypothesis class* $\mathcal{H}$ is the class of learning models, the *sample space* $\mathcal{S}$ is the space of datasets. The learning (algorithm) $A : \mathcal{S} \to \mathcal{H}$ is a mapping that assigns hypotheses to data. Generalization error is the discrepancy between the concept $c \in \mathcal{C}$ and the outcome $h = A(S_n) \in \mathcal{H}$. *Complexity (of learning model)* refers to an absolute size of hypothesis class $\mathcal{H}$, while *expressive power* refers to a size of $\mathcal{H}$ relative to concept class $\mathcal{C}$.

### 2.3 REGULARIZED EMPIRICAL RISK MINIMIZATION (LEARNING ALGORITHM)

As the learning algorithm, denoted $A$, we assume *(restricted) empirical risk minimization (RERM)* over abstract hypothesis class $\mathcal{H}_k$. Precisely, we consider a two-stage learning: First minimize the empirical risk, denoted $\hat{L}_n$, over abstract depth-$k$ networks $\mathcal{H}_k$, then approximate the minimizer in $\mathcal{H}_k$ by specific networks $\mathcal{H}_{\mathrm{imp}}$. In practice, trained networks obtained by optimization are not identical to exact empirical risk minimizers, but we assume they are identical for simplicity.

For the loss function $\ell : \mathcal{Y} \times \mathcal{Y} \to \mathbb{R}_{\geq 0}$, we assume boundedness and 1-Lipschitz continuity at a.e. second argument (as well as measurability): $0 \leq \ell(y', y) \leq b$, and $|\ell(y, y_o) - \ell(y', y_o)| \leq \beta_\ell |y - y'|$ a.e. $y_o \in \mathcal{Y}$. Examples include smoothed 0-1 loss in classification, or truncated and normalized squared loss in regression.

Given an i.i.d. sample $S_n := \{(X_i, Y_i)\}_{i=1}^n$ drawn from $P$, put the population risk as $L[h] := \mathbb{E}_{(X,Y) \sim P}[\ell(h(X), Y)]$, and the empirical risk as $\hat{L}(S_n)[h] := \frac{1}{n} \sum_{i=1}^n \ell(h(X_i), Y_i)$, respectively.

By $\mathrm{embed} : \mathcal{H}_k \to \mathcal{H}_{\mathrm{imp}}$, we denote an embedding operator from abstract network $\mathcal{H}_k$ to specific network $\mathcal{H}_{\mathrm{imp}}$. With this notation, the two-stage learning algorithm is formulated as

$$A(S_n) := \mathrm{embed}\left( \underset{f \in \mathcal{H}_k}{\arg\min}\, \hat{L}(S_n)[f] \right), \quad S_n \in (\mathcal{X} \times \mathcal{Y})^n. \tag{2}$$

### 2.4 GENERALIZATION ERROR

To formulate the generalization error, we introduce a standard machine learning framework as summarized in Fig. 2, which is composed of three classes and two processes: concept class $\mathcal{C}$, sample space $\mathcal{S}$, and hypothesis class $\mathcal{H}$. A concept $c$ generates a sample (or a dataset) $S_n \in \mathcal{S}$, and learning algorithm $A : \mathcal{S} \to \mathcal{H}$ mapping the sample $S_n$ to a hypothesis $\hat{h} \in \mathcal{H}$. The concept $c$ itself is assumed to be unknown, and so the learning algorithm $A$ is an estimator of $c$ under the constraint that it can only access $S_n$. Generalization error is the discrepancy between the concept $c \in \mathcal{C}$ and outcome $\hat{h} = A(S_n) \in \mathcal{H}$ of the learning algorithm.

In this study, the concept class $\mathcal{C}$ is a certain collection of measurable functions $c : \mathcal{X} \to \mathbb{R}$, the sample space $\mathcal{S}$ is the product space $(\mathcal{X} \times \mathcal{Y})^n$, and the hypothesis class is composed of two sub-classes: abstract depth-$k$ network $\mathcal{H}_k$ and specific network $\mathcal{H}_{\mathrm{imp}}$, and our two-stage learning algorithm $A$ is a composition of ERM $\mathcal{S} \to \mathcal{H}_k$ followed by embedding $\mathcal{H}_k \to \mathcal{H}_{\mathrm{imp}}$.

The generalization error refers to three related quantities: Population risk $L[\hat{h}]$, Excess risk $L[\hat{h}] - L_{\mathcal{C}}$, and Generalization gap $L[\hat{h}] - \hat{L}[\hat{h}]$. Here, $L_{\mathcal{C}} := \inf_{c \in C} L[c]$ is the infimum risk attainable over all the data generators in $\mathcal{C}$. In all the quantities, the focus is on the population risk $L[\hat{h}]$, but it is intractable because the data distribution $P$ is assumed to be unknown. Thus in the theoretical analysis we estimate the discrepancy from either the Bayes risk $L_{\mathcal{C}}$ or the training loss $\hat{L}[\hat{h}]$.

## 3 MAIN RESULTS

To state the generalization error bounds, the main theorem of this study, we additionally introduce two quantities: Rademacher complexity, and covering number. Further details are summarized in brief notes: Appendices G and H.

**Rademacher Complexity.** Let $\mathcal{H}$ be a separable set of real-valued measurable functions on a probability space $(\mathcal{X}, P)$ and let $S := \{X_1, \ldots, X_n\}$ be a sample drawn from distribution $P^n$. Let $\boldsymbol{\sigma} := \{\sigma_1, \ldots, \sigma_n\}$ be independent Rademacher variables (i.e. $\Pr\{\sigma_i = \pm 1\} = \frac{1}{2}$). The *empirical Rademacher complexity* $\hat{\mathfrak{R}}_S(\mathcal{H})$ of $\mathcal{H}$ on $S$ and the *(population) Rademacher complexity* $\mathfrak{R}_n(\mathcal{H})$ are respectively given by

$$\hat{\mathfrak{R}}_S(\mathcal{H}) = \mathbb{E}_{\boldsymbol{\sigma}}\left[\sup_{h \in \mathcal{H}} \frac{1}{n} \sum_{i=1}^n \sigma_i h(X_i)\right], \quad \text{and} \quad \mathfrak{R}_n(\mathcal{H}) := \mathbb{E}_{S \sim P^n}\left[\hat{\mathfrak{R}}_S(\mathcal{H})\right].$$

**Covering Number.** Given a (pseudo-)metric space $(\mathcal{M}, d)$ and positive number $\varepsilon > 0$, an $\varepsilon$-*cover* is a finite set $C \subset \mathcal{M}$ such that every $m \in \mathcal{M}$ lies within distance $\varepsilon$ of some $c \in C$. The covering number $N(\mathcal{M}, d, \varepsilon)$ is the size of the smallest such cover.

### 3.1 BIAS-VARIANCE DECOMPOSITION

Let $\varepsilon_{\mathrm{imp}}$ be the uniform approximation error of embedding from $\mathcal{H}_k \to \mathcal{H}_{\mathrm{imp}}$, i.e. $\varepsilon_{\mathrm{imp}} := \sup_{h \in \mathcal{H}_k} \|h - \mathrm{embed}[h]\|_\infty$, and let $\varepsilon_{\mathrm{model}}$ be the model bias of $\mathcal{H}_k$ relative to $\mathcal{C}$, i.e. $\varepsilon_{\mathrm{model}} := \inf_{h \in \mathcal{H}_k} L[h] - \inf_{c \in C} L[c]$.

Then, we have the following bias-variance decompositions of generalization error bound.

**Theorem 1** (Bias-Variance Decomposition for ERM over Depth-$k$ Networks). *With probability at least $1 - \delta$ over the draw of i.i.d. sample $S \sim P^n$, the excess risk and generalization gap are respectively decomposed as follows:*

$$L[\hat{h}] - L_{\mathcal{C}} \leq \varepsilon_{imp} + \varepsilon_{model} + 4\hat{\mathfrak{R}}_S(\mathcal{H}_k) + 2b\sqrt{\frac{2\log 1/\delta}{n}}, \tag{3}$$

$$L[\hat{h}] - \hat{L}[\hat{h}] \leq 2\varepsilon_{imp} + 2\hat{\mathfrak{R}}_S(\mathcal{H}_k) + b\sqrt{\frac{2\log 1/\delta}{n}}. \tag{4}$$

See Appendix B for the proof. Here, *bias* refers to approximation errors $\varepsilon_{\mathrm{imp}}$ and $\varepsilon_{\mathrm{model}}$, and *variance* refers to estimation error $\hat{\mathfrak{R}}_S(\mathcal{H}_k)$. The bias-variance decomposition is carefully designed so that the variance term depends only on the abstract network $\mathcal{H}_k$ and does not depend on specific network $\mathcal{H}_{\mathrm{imp}}$. Thus for variance analysis, we can concentrate on abstract network $\mathcal{H}_k$, independent of network architectures/implementations.

### 3.2 HIDDEN-OUTPUT DECOMPOSITION OF RADEMACHER COMPLEXITY

We can further decompose the variance term into hidden and output layers as follows. Given a finite sample $S := \{X_i\}_{i=1}^n$, let $d_S$ denote the *empirical (pseudo-)metric* of maps $f, g \in C(X, X)$ defined by $d_S(f, g) := \left(\frac{1}{n}\sum_{i=1}^n d(f(X_i), g(X_i))^2\right)^{1/2}$. Similarly, let $\|\cdot\|_S$ denote the *empirical (pseudo-)norm* of function $h \in C(X)$ defined by $\|h\|_S := \left(\frac{1}{n}\sum_{i=1}^n |h(X_i)|^2\right)^{1/2}$.

**Theorem 2** (Hidden-Output Decomposition of Rademacher Complexity for Depth-$k$ Network). *Assume that $H \subset C(\mathcal{X})$ is uniformly $L$-Lipschitz, and $F \subset C(\mathcal{X}, \mathcal{X})$ is totally-bounded with identity element $\mathrm{id}_{\mathcal{X}}$. Write $B_k := B(k, F)$ for short. The Rademacher complexity of composition class $\mathcal{H}_k := H \circ B_k$ is decomposed into the sum of the complexities of output class $H$ and hidden class $F$ as follows:*

$$\hat{\mathfrak{R}}_S(\mathcal{H}_k) \leq \hat{\mathfrak{R}}_S(H) + \frac{12L}{\sqrt{n}} \int_0^{\mathrm{diam}(B_k)} \sqrt{\log N(B_k, d_S, \varepsilon)}d\varepsilon. \tag{5}$$

*Here, $\mathrm{diam}(B_k) := \sup_{f \in B_k} d_S(f, \mathrm{id}_{\mathcal{X}})/2$ denotes the diameter of $B_k$ in $d_S$.*

See Appendix C for the proof. Based on this decomposition, we can analyze the effects of the output and hidden layers on estimation error separately. Particularly, the effect of hidden layer $F$ is much clearer than the composite form $H \circ F$.

*Remark* 1. While upper-bounding Rademacher complexity via covering numbers is classically known as Dudley's entropy integral (Theorem 8), to the best of our knowledge our bound is novel for two reasons. First, the second term is the entropy integral of $\mathcal{X}$-valued maps $F \subset C(\mathcal{X}, \mathcal{X})$, for which Rademacher complexity cannot be defined. By the construction, the Rademacher complexity can be defined only for real-valued functions such as $H \subset C(\mathcal{X})$. Thus prior deep learning studies based on the Rademacher complexity analysis have largely avoided such a formulation as "$\hat{\mathfrak{R}}_n(F)$".

Second, the first term retains the form of a Rademacher complexity rather than an entropy integral, which is convenient because it leaves open computational routes other than covering-number bounds. The proof of this mixed bound is somewhat technical. A simpler (but looser) derivation replaces the Rademacher complexity of $\mathcal{H}_k$ by the entropy integral over the whole class and then factors the covering number of $\mathcal{H}_k$ into that of the output class $H$ and of the depth-$k$ hidden maps $B_k$, yielding a two-entropy-integral estimate:

$$\hat{\mathfrak{R}}_S(\mathcal{H}_k) \leq \frac{12}{\sqrt{n}} \int_0^{\mathrm{diam}(\mathcal{H}_k)} \left[ \sqrt{\log N(H, \|\cdot\|_S, \varepsilon/2)} + \sqrt{\log N(B_k, d_S, \varepsilon/2L)} \right] \mathrm{d}\varepsilon. \quad (6)$$

In Appendix D, we have supplemented the simpler proof.

## 4 Growth Rates Analysis

In this section we quantify how the covering number of the depth-$k$ network, $N(B(k,F), d_\infty, \varepsilon)$, grows with $k$. Through Dudley's entropy integral (Theorem 2), this growth governs the depth dependence of the variance term in our generalization bounds (Theorem 1): exponential growth of $N(B(k,F), d_\infty, \varepsilon)$ yields polynomial growth of the variance, whereas polynomial growth of $N$ yields only logarithmic variance growth. We therefore seek implementation-agnostic conditions that force saturation (no growth), polynomial growth, or (super/double-)exponential growth in $k$.

Technically, we focus on the covering number $N$ in $d_\infty$, rather than in $d_S$. Here $N(\bullet, d_\infty, \varepsilon)$ upper-bounds $N(\bullet, d_S, \varepsilon)$ because $d_S \leq d_\infty$. Besides, we focus on the case where generator $F$ is infinite and compact, namely *compactly generated* semigroup $\langle F \rangle$, since typical neural network parameters form finite-dimensional manifolds, thus the induced semigroups are infinitely generated in general.

Below, we only summarize conditions. The proofs and examples are provided in Appendix E. Arzelà–Ascoli theorems are summarized in Appendix I.

### 4.1 Overview of Conditions

Conditions P1 and P2 explain variance at most logarithmic in $k$; Conditions E1, E2 and E3 identify polynomial or worse variance growth. The relationships between each condition can be organized as shown in Table 1

Table 1: Simplified look-up table mapping from conditions on space $\mathcal{X}$ and generator $F$ to growth rates (saturate, polynomial, or (at least) exponential). Note that this is simplified and incomplete; for example, the trichotomy $\mathrm{Lip}\, F \lesseqgtr 1$ is easy to check but cannot completely classify the conditions.

|  | $\mathrm{Lip}\, F < 1$ (contractive) | $\mathrm{Lip}\, F = 1$ (isometric) | $\mathrm{Lip}\, F > 1$ (expansive) |
|---|---|---|---|
| $\mathcal{X}$ compact | saturate (P1) | saturate (P1) | (poly, P2) or (exp, E2) |
| $\mathcal{X}$ non-compact | (saturate, P1') | (poly, P2) or (exp, E1) | (super-exp, E3) |

When the underlying space $\mathcal{X}$ is *compact* and the generating system $F$ is *non-expansive*, the Arzelà–Ascoli theorem implies that the generated semigroup is (relatively) compact; hence the covering numbers saturate at a depth-independent constant (Condition P1). Therefore, for the covering numbers to grow with depth, one needs either *non-compactness* or *non-contracting* behavior.

Indeed, when at least one of these two compact/non-expansive requirements fails—namely, in the cases *(non-compact + isometric)* and *(compact + expanding)*—one can show *exponential growth* under suitable additional assumptions (Conditions E1 and E2).

For *finitely generated groups*, the *polynomial* growth is attained when and only when the group is *virtually nilpotent* (Gromov, 1981); and *exponential* growth is attained when the group is *free*.

Similarly, for *compactly generated semigroups*, at most *polynomial* growth is attained when the semigroup embeds into a *nilpotent Lie group* such as an abelian group and the Heisenberg group (Condition P2); and at least *exponential* growth is attained when the semigroup contains a *free group* with satisfying appropriate *separation conditions* (Conditions E1 and E2). Moreover, one can even attain *super-* or *double-exponential* growth when the graph exhibits *uniformly-expanding* (Condition E3).

On the other hand, unlike in the case of finitely generated groups, the growth rate cannot be determined solely from the *independent* complexity (or simplicity) of the base space $\mathcal{X}$ or of the generating system $F$. For example, although both Examples 3 and 11 treat the Cantor set (a complex space) and a free action (a complex action), the former saturates to a depth-independent constant (i.e., grows polynomially with degree 0), whereas the latter grows exponentially. The reason is that merely having a free-group or tree-like structure in the generators is insufficient: if the *separation* condition fails and distances between deep nodes *contract*, then by Arzelà–Ascoli (Condition P1) the dependence collapses to a constant rather than becoming exponential.

### 4.2 Sufficient Conditions for Polynomial Upperbounds of Covering Numbers (Logarithmic Growth of Entropies)

**Condition P1 (Equicontinuous on Compact Saturates).** If the state space $\mathcal{X}$ is compact and the generated semigroup $\langle F \rangle$ is precompact/equicontinuous—e.g., if maps are uniformly Lipschitz or non-expanding—then Arzelà–Ascoli implies $\langle F \rangle$ is compact and the covering numbers of $B(k, F)$ saturate: they do not depend on $k$ at any fixed $\varepsilon > 0$.

*Examples.* (Example 1) Rotations on a circle, $\mathcal{X} = \mathbb{S}^1$ and $F \subset \mathrm{Iso}(\mathbb{S}^1)$, yield saturation. (Example 2) Finite isometry groups on finite set $\mathcal{X}$: growth is trivially bounded. (Example 3) A contractive iterated-function system on $[0, 1]$ producing the Cantor set: once depth exceeds an $\varepsilon$-dependent threshold, $N(B(k, F), \varepsilon)$ is controlled by the Cantor attractor and no longer grows with $k$.

**Condition P2 (Nilpotent Grows Polynomially).** Suppose $\langle F \rangle$ embeds bi-Lipschitzly into a connected, simply connected nilpotent Lie group $H$ that acts on $\mathcal{X}$. Then there exists a constant $D$ (the Guivarc'h–Bass homogeneous dimension of $H$) such that

$$N\big(B(k, F), d_\infty, \varepsilon\big) \lesssim \Big(1 + \tfrac{k}{\varepsilon}\Big)^D.$$

Thus the hidden-layer contribution to the variance grows at most logarithmically with $k$.

*Examples.* (Example 4) Translations on Euclidean space $\mathcal{X} = \mathbb{R}^d$ (non-compact, abelian, isometric): $O((k/\varepsilon)^D), D \leq d$. (Example 5) Shear on torus $\mathcal{X} = \mathbb{T}^2$ (compact, abelian, expanding): $= 1 + k$ (independent of $\varepsilon < 1$). (Example 6) Discrete Heisenberg group actions on torus $\mathcal{X} = \mathbb{T}^3$ (compact, nilpotent, expanding): $\Theta(1 + k)^D, D = 4$ (independent of $\varepsilon < \varepsilon_0$). (Example 7) Upper-Triangular group bi-Lipschitz actions on $\mathcal{X} = \mathbb{R}^d$ (non-compact, nilpotent, expanding): $\Theta(1 + k/\varepsilon)^D, D = \frac{d(d-1)(d+1)}{6}$.

### 4.3 Sufficient Conditions for Exponential Lowerbounds of Covering Numbers (Polynomial Growth for Entropies)

**Condition E1 (Free Semigroup with Point Separation Grows Exponentially)** If the semigroup generated by $F = \{f_1, \ldots, f_r\}$ is free at every length and there exist $x_* \in \mathcal{X}$ and $\delta > 0$ such that for all distinct words $u, v$ of the same length $d(f_u(x_*), f_v(x_*)) \geq \delta$, then for every $\varepsilon < \delta/2$ and every depth $k$,

$$N\big(B(k, F), d_\infty, \varepsilon\big) \geq r^k.$$

Separation at a single probe transfers the exponential word count to the covering number.

*Examples.* (Example 8) (Free group, standard word metric; non-compact, isometric) Let $\mathcal{X} = F_r$ (non-compact) with standard word metric and $F = \{L_{a_i}\}$ (isometric); at the basepoint $e$, distinct words of the same length are $\geq 2$ apart, so for $\varepsilon < 1$, $N(B(k,F), d_\infty, \varepsilon) \geq r^k$, even though $d_\infty(L_u, L_v) = \infty$ for $u \neq v$. Example 9 (Free group, bi-invariant metric; non-compact, isometric) Let $\mathcal{X} = F_r$ with a conjugacy-invariant (bi-invariant) word metric and $F = \{L_{a_i}\}$ (isometric); then $d_\infty(L_g, L_h) = d(e, g^{-1}h) < \infty$ and at the basepoint $e$, distinct words satisfy $d \geq 1$, so for $\varepsilon < 1/2$, $N(B(k,F), d_\infty, \varepsilon) \geq r^k$.

**Condition E2 (Ping–Pong Grows Exponentially)** Suppose there are pairwise separated "chambers" $U_i$ and anchors $a_i$ with $f_i$ mapping $U_i$ surjectively and expansively across chambers while collapsing $\mathcal{X} \setminus U_i$ near $a_i$, and $\min_{i \neq j} d(a_i, a_j) > 0$; then for some $\delta > 0$, every $\varepsilon < \delta/2$ and depth $k$ satisfy

$$N\big(B(k,F), d_\infty, \varepsilon\big) \geq r^k.$$

The rightmost mismatch in two words always yields a uniform output gap.

*Examples.* (Example 10) (interval, piecewise-linear expand/reset; compact, expansive) Let $\mathcal{X} = [0,1]$ (compact) and $F = \{f_0, f_1\}$ be PL maps that expand (slope $> 1$) on their own subintervals and reset to constants elsewhere; the ping–pong conditions give a uniform $\delta = 1/3 - 2\eta > 0$, hence for $\varepsilon < \delta/2$, $N(B(k,F), d_\infty, \varepsilon) \geq 2^k$. (Example 11) (Subshift ping–pong; compact, expansive). Let $\mathcal{X} = \mathcal{A}^{\mathbb{N}}$ (compact Cantor space with ultrametric $d_\theta$) and define $f_a$ to act as the shift on the cylinder $[\![a]\!]$ (expansive) and as the constant $\bar{a}$ off $[\![a]\!]$; chambers and anchors are uniformly separated ($\delta = 1$), so for $\varepsilon < 1/2$, $N(B(k,F), d_\infty, \varepsilon) \geq r^k$.

**Condition E3 (Uniform Expansion for Super/Double-Exponential).** Assume there is a *reset* map $r$ sending all of $\mathcal{X}$ into a compact $k$, a uniformly expanding map $A$ on $k$ with co-Lipschitz constant $\lambda > 1$, and a compact family of 1-Lipschitz *writers* $G \subset C(K, K)$. Then, for words of the form $A \circ g_k \circ A \circ \cdots \circ A \circ g_1 \circ r$,

$$N\big(B(Ck + O(1), F), \varepsilon\big) \gtrsim \prod_{j=1}^k N\Big(G, \varepsilon/\lambda^{k-j+1}\Big),$$

which amplifies layerwise entropy multiplicatively. Consequences include super-exponential growth when $G$ sits on a $D$-dimensional $C^0$-submanifold (giving $\log N \gtrsim k^2$) and double-exponential growth when $G$ is a Hölder ball on a $d$-dimensional manifold (giving $\log N \gtrsim \lambda^{(d/\alpha)k}$).

*Example.* (Example 12) On $\mathcal{X} = \mathbb{R}^d$ (with a bounded base metric), one can take a radial reset $r$, a near-linear expansion $A$ on $K = B(0,1)$, and writers $G = \{\mathrm{id} + u \mid \|u\|_\infty \leq 1, [u]_{C^\alpha} \leq 1\}$, which realizes the double-exponential case.

## 5 Optimal Depth

As an application of our general results, we estimate the *optimal depth* by balancing the approximation and estimation errors. As the bias-variance decomposition suggested, the estimation term (variance), denoted $\mathsf{var}(k, n)$, is an intrinsic quantity depending solely on the hypothesis class $\mathcal{H}_k$ itself, whereas the approximation term (bias), denoted $\mathsf{bias}(k)$, is an extrinsic quantity depending not only on $\mathcal{H}_k$ but also on the concept class $\mathcal{C}$. Thus the same hypothesis class $\mathcal{H}_k$ can yield different approximation error rate depending on the choice of concept class $\mathcal{C}$.

Here we analyze regimes where the approximation error decays with depth $k$ either exponentially or polynomially:

$$\mathsf{bias}(k) = \exp(-\alpha k) \text{ (Exponential decay)}, \quad \text{or} \quad k^{-\beta} \text{ (Polynomial decay)}$$

with parameters $\alpha, \beta > 0$, and where the covering number (governing estimation error) grows polynomially or exponentially, yielding estimation error bounds of root-log or root-polynomial growths:

$$\mathsf{var}(k, n) = \sqrt{\log(k)/n} \text{ (root-Logarithmic growth)}, \quad \text{or} \quad \sqrt{k^\gamma/n} \text{ (root-Polynomial growth)}$$

with parameter $\gamma > 0$. Thus by combining the *two* bias decay rates with the *two* variance growth rates, we obtain the $(2 \times 2 =)$ *four regimes: EL, EP, PL, and PP.*

Table 2: Example of the optimal depth balancing bias and variance

| Regime | bias$(k)$ | var$(k,n)$ | Optimal depth $k^*$ | Optimal gen$(k^*,n)$ |
|---|---|---|---|---|
| EL | $\exp(-\alpha k)$ | $\sqrt{\log(k)/n}$ | $\log(n)/(2\alpha) + \tilde{o}(1)$ | $\asymp \sqrt{\log(\log(n))/n}$ |
| EP | $\exp(-\alpha k)$ | $\sqrt{k^\gamma/n}$ | $\log(n)/(2\alpha) + \tilde{O}(1)$ | $\asymp \sqrt{(\log n)^\gamma/(2\alpha)^\gamma n}$ |
| PL | $k^{-\beta}$ | $\sqrt{\log(k)/n}$ | $\sim (2\beta n/\log(2\beta n))^{1/(2\beta)}$ | $\asymp \sqrt{\log(n)/(2\beta n)}$ |
| PP | $k^{-\beta}$ | $\sqrt{k^\gamma/n}$ | $\asymp n^{1/(2\beta+\gamma)}$ | $\asymp n^{-\beta/(2\beta+\gamma)}$ |

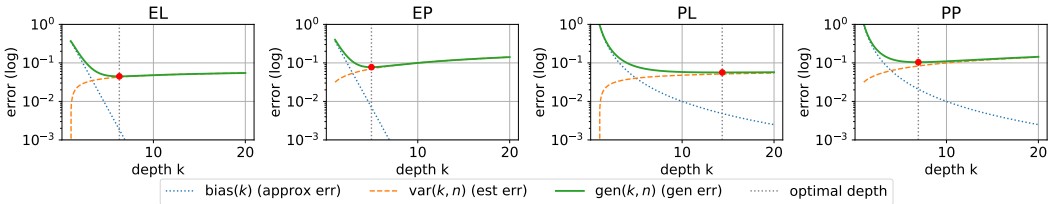

Figure 3: Typical examples of approximation error and estimation error ($\alpha = 1.0, \beta = 2.0, \gamma = 1.0$)

By equating leading terms, the optimal depths are estimated as in Table 2. Typical examples are also visualized in Fig. 3. The details of calculations are described in Appendix F.

In every regime, the optimal depth $k^*$ grows with training sample size $n$: larger datasets require deeper networks. Precisely, the optimized generalization error increases (= gets worse) in the order: EL $\lesssim$ EP $\lesssim$ PL $\lesssim$ PP, and the exp-decay bias yields a shallower $O(\log n)$-depth, while poly-decay bias yields a deeper $O(\text{poly } n)$-depth. Therefore, in general, the approximation efficiency has a stronger influence on the generalization performance than the model complexity.

## 6 EXAMPLES

Here, we discuss specific examples that fall under the four regimes.

### 6.1 CONTRACTIVE TEACHER-STUDENT SETTING (EL)

A teacher-student setting refers to the scenario where the hypothesis class (student) and the concept class (teacher) share the same compositional structure. Let the depth-$k$ student class be $\mathcal{H}_k := H \circ B(k, F)$ and take the concept class to be the infinitely deep limit $\mathcal{C} := \mathcal{H}_\infty := H \circ B(\infty, F)$. Assume the input space $\mathcal{X}$ is bounded with $\text{diam}(\mathcal{X}) = D < \infty$; the output layer $H$ is 1-Lipschitz; and the intermediate-layer semigroup $F \subset C(\mathcal{X}, \mathcal{X})$ is *contractive*, i.e., there exists $\lambda \in (0, 1)$ such that $\text{Lip}(f) \leq \lambda$ for all $f \in F$. Then any teacher $c \in \mathcal{H}_\infty$ can be written as $c = h \circ u \circ v$ with $h \in H$, $u \in B(k, F)$, and $v \in B(\ell, F)$ for some $\ell$. Truncating the tail yields a depth-$k$ student $h \circ u \in \mathcal{H}_k$ that approximates $c$ with error

$$\|h \circ u \circ v - h \circ u\|_\infty \leq \text{Lip}(h)\,\text{Lip}(u)d_\infty(v, \text{id}) \leq 1 \cdot \lambda^k \cdot D = D\lambda^k,$$

since $\text{Lip}(u) \leq \lambda^k$ and $d_\infty(v, \text{id}) \leq D$. Hence the approximation error decays exponentially:

$$\sup_{c \in \mathcal{H}_\infty} \inf_{h \in \mathcal{H}_k} \|c - h\|_\infty \leq D\lambda^k.$$

Moreover, the contraction assumption implies the logarithmic variance growth due to Condition P1. Therefore, a teacher-student setting with contractive assumption falls under the EL regime.

### 6.2 NEURAL OPERATOR (EL)

Furuya et al. (2025) investigated the approximation error of Neural Operators (NOs) that learn the solution operator of nonlinear parabolic PDEs on a bounded domain $\mathcal{X} \subset \mathbb{R}^d$. By aligning a single hidden layer of the NO with one step of the Picard iteration for the PDE, a textbook iterative arguments for DE solving, they show that the approximation error decays at a *sub-exponential* rate

$\exp O(-\sqrt{k})$ in the network depth $k$. Since Picard iteration is *contractive* and, moreover, generated by a *single* operator, the resulting NO forms a *commutative contractive semigroup*, implying that the covering numbers grow only polynomially with depth (Conditions P1 and P2). The function space for approximation is a *mixed Lebesgue space* $L_t^r L_x^s$, which we can regard as the concept class $\mathcal{C}$.

This NO setting can be viewed as a concrete instance of the teacher-student framework discussed earlier: both the hypothesis class and the concept class share a hierarchical (iterative) structure, enabling efficient depth-driven approximation. It thus exemplifies the *PL regime*, in which deep learning is particularly advantageous. Together with the previous example, it is suggested that deep learning is most effective when the underlying data space, namely the concept class $\mathcal{C}$, possesses a hierarchical structure such as that induced by deep compositions or differential equations.

### 6.3 PARTIALLY-FREE TEACHER-STUDENT SETTING (EP)

As mentioned in the EL example (and Condition P1), networks generated by a contractive semigroup on a compact space are precompact, so their covering numbers do not grow with depth. While slow growth of estimation error is preferable in practice, we present one artificial construction where estimation complexity grows exponentially in depth, even though approximation error still decays exponentially. The basic idea is again in a teacher-student setting to make the hypothesis class $\mathcal{H}_k$ deliberately *larger* (redundant) than the concept class $\mathcal{C}$.

Let the input space $\mathcal{X}$ be bounded with $\mathrm{diam}(\mathcal{X}) = D < \infty$. Assume the output layer $H$ is 1-Lipschitz. For hidden layer maps, take the union $F := S \cup T$ of a *contractive* semigroup $S \subset C(\mathcal{X}, \mathcal{X})$ with $\sup_{s \in S} \mathrm{Lip}(s) =: \lambda < 1$ and a *free* part $T$ with finite generators whose generated semigroup $\langle T \rangle := B(\infty, T)$ is $\rho$-separated in the uniform metric: there exists $\rho > 0$ such that $d_\infty(f, g) \geq \rho$ for all distinct $f, g \in \langle T \rangle$. This implies a covering-number lower bound $N(B(k, T), \varepsilon) \gtrsim |T|^k$ for any $\varepsilon < \rho/2$.

Define the concept class by the contractive part only, $\mathcal{C} := H \circ B(\infty, S)$, and the depth-$k$ hypothesis class by the full union, $\mathcal{H}_k := H \circ B(k, S \cup T)$. Then we are in the *EP regime*: the approximation error decays exponentially by the same truncation argument as before:

$$\sup_{c \in H \circ B(\infty, S)} \inf_{h \in H \circ B(k, S \cup T)} \|c - h\|_\infty \leq D\lambda^k.$$

On the other hand, the covering numbers of the hidden maps grow at least exponentially with depth, because (Condition E1 holds, or)

$$N\big(B(k, S \cup T), \varepsilon\big) \geq N\big(B(k, T), \varepsilon\big) \gtrsim |T|^k \quad (\varepsilon < \rho/2),$$

so the estimation term exhibits exponential dependence on $k$. Such an example can be made fully concrete by taking $\mathcal{X}$ to be the Hilbert cube $[0, 1]^\mathbb{N}$ equipped with $\ell^\infty$-norm and constructing explicit choices of $S$ (contractive) and $T$ (finite, $\rho$-separated free generators) on this space.

### 6.4 RELU NETWORKS IN HÖLDER-SMOOTH SPACE (PP/PL)

A canonical setting where approximation error decays only at a *polynomial* rate is given by *Jackson-type* bounds for *Hölder* $C^s([0, 1]^d)$, *Sobolev* $W^{s,p}([0, 1]^d)$, and *Besov* $B_q^{s,p}([0, 1]^d)$ spaces: For such a function $f$ in these spaces, the best $m$-parameter approximation achieves order $O(m^{-s/d})$, with matching lower bounds $\Omega(m^{-s/d})$ under very mild assumptions (DeVore et al., 1989). Thus, exponentially fast approximation cannot be expected in these spaces.

A line of expressive power analysis of ReLU networks initiated by Yarotsky (Yarotsky, 2017; 2018; Yarotsky & Zhevnerchuk, 2020; Siegel, 2023; Yang & He, 2024) shows that deep ReLU networks can attain the so-called *super-convergence*, or surpass the Jackson's rates, by combining piecewise-polynomial approximation with *bit-extraction*, a highly compressed, discontinuous encoding technique (from function to parameter); the speedup hinges on violating the regularity assumptions underlying the Jackson-type lower bounds, yet the decay remains polynomial rather than exponential. Although a rigorous estimation error analysis was not provided by the authors, bit-extraction is a Cantor attractor, which may yield both a polynomial growth (Example 3) and an exponential growth (Example 11) of covering numbers. Thus, we expect the settings where the hypothesis class is ReLU network and concept class is Hölder/Sobolev/Besov spaces fall under *PP/PL regime*.

Apart from the Jackson's regime, estimation error for ReLU networks has been shown to grow *polynomially* in depth $k$ via VC-dimension arguments (Bartlett et al., 2019) and compression-based generalization bounds (Arora et al., 2018; Suzuki et al., 2020; Lotfi et al., 2022). Putting these observations together: taking the concept class $\mathcal{C}$ as Hölder/Sobolev/Besov, and the hypothesis class $\mathcal{H}_k$ as depth-$k$ ReLU networks yields the *PP regime*.

We remark that Suzuki (2019) investigated both approximation and estimation error rates for ReLU networks in both Besov and mized-smooth Besov spaces, and obtained exactly the *PP regime*: polynomial bias decay with polynomial variance growth.

## 6.5 ReLU Networks in Hierarchical Class (PL)

Schmidt-Hieber (2020) developed a hierarchical class, named *composite function class $G$*, obtained by compositions of Hölder-smooth maps and showed that deep ReLU networks achieve the minimax-optimal rate. Their argument bounds the covering numbers of deep ReLU classes, yielding estimation terms that increase only *logarithmically* in depth $k$. On the approximation side, they obtain a bound with two terms: an *exponentially* decaying term in depth $k$ (from compositional structure) plus a *polynomial* Jackson's rate term in the number of parameters $m$. While this is not purely a polynomial decay in depth, the overall picture fits within a *PL regime*.

## 7 Conclusion

We developed a unified framework for analyzing the depth dependence of generalization error encompassing a variety of network architectures by splitting deep learning models into abstract state transitions composed with concrete implementations. We analyzed generalization through a bias-variance decomposition in which the variance term depends only on the state-transition semigroup. We characterized when the covering numbers of semigroups grow polynomialy or exponentially with depth, yielding the root-logarithmic or root-polynomial variance growth. We combined these results with exponential or polynomial bias decay to obtain four bias-variance trade-off regimes (EL, EP, PL, PP) and their optimal depths. The optimal depth $k^*$ is typically greater than one, which constitutes a rigorous form of *depth supremacy*. Among the four regimes, depth performs the best in EL, where bias decays Exponentially while variance grows only Logarithmically. Hierarchical or iterative concept classes $\mathcal{C}$ such as Neural ODEs, diffusion/score models, and chain-of-thought reasoning models naturally realize the EL regime. On the other hand, classical Hölder/Sobolev/Besov spaces tend to yield PL/PP regimes and make depth supremacy harder to establish. Our analysis highlights *compactly-generated semigroups* as a unifying mathematical object and connects depth-generalization to ideas from coarse geometry and dynamical systems. We hope this bridge will motivate further interaction between modern mathematics and the theory of deep learning.

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

# A  LITERATURE OVERVIEW

**Depth separation and expressivity.** A classical line of work shows that modest increases in depth can yield exponential representational advantages. Eldan & Shamir (2016) proved a three-vs-two-layer separation for a simple radial function, requiring exponential width for any depth-2 approximant, while Telgarsky (2016) established families exhibiting exponential gaps between networks of depth $O(k^3)$ and $O(k)$ with semi-algebraic gates (including ReLU) and provided constructive hard instances; subsequent work extended separations beyond radial constructions. These results clarify when *expressivity* favors depth, but do not by themselves pin down estimation behavior.

**Generalization via capacity control.** Combinatorial analyses give nearly tight VC/pseudodimension bounds for piecewise-linear networks, scaling roughly linearly in depth for fixed width/weights, providing a baseline picture of depth in classical uniform-convergence frameworks (Bartlett et al., 2019). Norm- and margin-based approaches bound estimation error through products of layer norms (path/spectral) (Neyshabur et al., 2015; Bartlett et al., 2017), sometimes yielding size-independent or gently depth-dependent bounds under additional structure (e.g., margin normalization). These strands highlight multiple possible depth dependencies—linear, polynomial, or even milder—depending on how complexity is measured.

**Rademacher/covering and size-independent bounds.** A complementary thread controls depth via data-dependent complexities (Rademacher, covering). Golowich et al. (2018; 2020) obtained bounds that (under norm constraints) improve the depth dependence and can be independent of width and depth in certain regimes; later refinements further reduced explicit depth factors. These works show how the estimation side may be decoupled from naively counted parameters and instead tied to geometric quantities of the hypothesis class.

**Compression and PAC-Bayes.** A productive viewpoint explains generalization via compressibility: if a trained network admits a succinct reparametrization, one can transfer that compression into generalization guarantees. Arora et al. (2018) formalized this link and demonstrated strong bounds in practice; follow-ups convert compression bounds to the original (non-compressed) networks and sharpen them via PAC-Bayes with subspace quantization, yielding state-of-the-art nonvacuous estimates (Suzuki et al., 2020; Lotfi et al., 2022). While depth typically enters these bounds through compressibility or margin quantities, the methodology is agnostic to architecture details.

**Nonparametric regression with deep networks.** Another large body of work analyzes approximation–estimation trade-offs of ReLU networks on smoothness classes. Schmidt-Hieber (2020) showed near-optimal rates in nonparametric regression, with depth playing an essential role; Suzuki (2019) established optimal adaptivity on (mixed) Besov spaces and improvements over linear/kernel baselines; Nakada & Imaizumi (2020) tied generalization to intrinsic (Minkowski) dimension; and Imaizumi & Fukumizu (2019; 2022) identified regimes with singularities where DNNs are minimax-superior to traditional estimators. Recent approximation results (e.g., optimal Sobolev/Besov rates) further sharpen the expressivity side (Yarotsky, 2017; 2018; Yarotsky & Zhevnerchuk, 2020; Siegel & Xu, 2022; Siegel, 2023; Yang & He, 2024). These analyses, however, are typically architecture-specific (ReLU feedforward) and hinge on smoothness assumptions.

**Iterative/hierarchical models and continuous depth.** Many modern systems are naturally modeled as *compositions* or *flows*—precisely the setting of our state-transition abstraction. Neural ODEs (Chen et al., 2018) as well as Neural Operators (Kovachki et al., 2021) treat depth as continuous time evolution; diffusion/score-based models (Shen et al., 2025) implement long iterative refinement; and chain-of-thought (Wei et al., 2022) prompting in LLMs explicitly unfolds multi-step reasoning. These families motivate studying depth-dependent generalization at the level of abstract state transitions rather than fixed architectures.

**This Study.** Relative to these threads, this study is *implementation-agnostic*: instead of parameterizing a specific architecture, we analyze *state-transition semigroups on metric spaces*, derive depth dependence of the *variance* via covering/Rademacher complexity of word balls, give conditions for *polynomial/logarithmic* growth, and couple them with *exponential/polynomial* bias decay to compute *optimal depth* across four regimes. This yields a unified lens for when and why *depth supremacy* (optimal $k^* > 1$) emerges—particularly in iterative/hierarchical settings suggested above.

## B   PROOF OF THEOREM 1

Here we show a slightly generalized version of Theorem 1 in the main text. Since the abstract model class, denoted $\mathcal{H}_k$ in the main text, need not be a state-transition model $H \circ B(k, F)$, we simply write $\mathcal{H}_{\mathrm{abs}}$ instead. The main claim of this theorem is that in the setting where a hypothesis class $\mathcal{H}_{\mathrm{imp}}$ with a specific implementation, such as ReLU, approximates a hypothesis class $\mathcal{H}_{\mathrm{abs}}$ without an implementation, like a state-transition model, the generalization error bound can be decomposed into bias and variance so that the bias depends on both $\mathcal{H}_{\mathrm{imp}}$ and $\mathcal{H}_{\mathrm{abs}}$, while the variance depends only on $\mathcal{H}_{\mathrm{abs}}$ (and not on $\mathcal{H}_{\mathrm{imp}}$).

**Assumptions for the high probability bounds:**

- Let $\mathcal{X}$ be a measurable space, and let $T(\mathcal{X})$ be a topological space of real-valued measurable functions $h : \mathcal{X} \to \mathbb{R}$
- Let $\mathcal{Y}$ be a metric space
- Let $\mathcal{Z} := \mathcal{X} \times \mathcal{Y}$, and let $(\mathcal{Z}, P)$ be a probability space
- Let $S_n := (Z_i)_{i \in [n]} \sim P$ be an i.i.d. sample
- Let $\mathcal{H}_{\mathrm{abs}}$ be a separable subspace of $T(\mathcal{X})$
- Let $\ell : \mathcal{Y} \times \mathcal{Y} \to [0, b]$ be a bounded non-negative measurable function, and assume that the first argument is $\beta_\ell$-Lipschitz uniformly in the second argument
- Write $\ell \cdot h(x, y) := \ell(h(x), y)$, and $\ell \cdot \mathcal{H}_{\mathrm{abs}} := \{\ell \cdot h(x, y) \mid h \in \mathcal{H}_{\mathrm{abs}}\}$
- Let $L[h] := \mathbb{E}_{(X,Y) \sim P}[\ell \cdot h(X, Y)]$ and $\hat{L}(S_n)[h] := \frac{1}{n} \sum_{i=1}^n \ell \cdot h(X_i, Y_i)$

$\implies$ (the Lipschitz assumption yields $\mathfrak{R}(\ell \cdot \mathcal{H}_{\mathrm{abs}}) \leq \beta_\ell \mathfrak{R}(\mathcal{H}_{\mathrm{abs}})$ and) Theorem 6 holds at Eqs. (13) and (15).

**Assumptions for minimization:**

- Let $\mathcal{C}$ be a subspace of $T(\mathcal{X})$
- Assume $L$ is bounded below over $\mathcal{C}$
- Assume $\mathcal{H}_{\mathrm{abs}}$ is compact in $T(\mathcal{X})$, and both $L$ and $\hat{L}(S_n)$ are (lower)semicontinuous on $\mathcal{H}_{\mathrm{abs}}$
- Write $L_{\mathcal{C}} := \inf_{c \in C} L[c]$, $L_{\mathcal{H}_{\mathrm{abs}}} := \inf_{h \in \mathcal{H}_{\mathrm{abs}}} L[h]$, and $\varepsilon_{\mathrm{model}} := L_{\mathcal{H}_{\mathrm{abs}}} - L_{\mathcal{C}}$
- Let $f^\star := \arg\min_{f \in \mathcal{H}_{\mathrm{abs}}} L[f]$ and $\hat{f} := \arg\min_{f \in \mathcal{H}_{\mathrm{abs}}} \hat{L}(S_n)[f]$

$\implies$ Exist the infima $L_{\mathcal{C}}$ and $L_{\mathcal{H}_{\mathrm{abs}}}$, error $\varepsilon_{\mathrm{model}}$, and the minimizers $f^\star$ and $\hat{f}$.

**Assumptions for implementation error bound:**

- Assume $(T(\mathcal{X}), d_T)$ is a (pseudo-)metric space of real-valued measurable functions $h : \mathcal{X} \to \mathbb{R}$
- Let $\mathcal{H}_{\mathrm{imp}}$ be a compact subspace in $T(\mathcal{X})$
- Assume $\varepsilon_{\mathrm{imp}} := \sup_{f \in \mathcal{H}_{\mathrm{abs}}} \inf_{h \in \mathcal{H}_{\mathrm{imp}}} d_T(h, f) < \infty$
- Define $\mathrm{embed} : \mathcal{H}_{\mathrm{abs}} \to \mathcal{H}_{\mathrm{imp}}$ by $\mathrm{embed}[f] := \arg\min_{h \in \mathcal{H}_{\mathrm{imp}}} d_T(h, f)$
- Assume both $L$ and $\hat{L}(S_n)$ are $\beta_L$- and $\beta_{\hat{L}(S_n)}$ Lipschitz over $\mathcal{H}_{\mathrm{abs}}$, respectively

$\implies$ embed is well-defined, and for any $f \in \mathcal{H}_{\mathrm{abs}}$ and $h := \mathrm{embed}(f)$, $|L[h] - L[f]| \leq \beta_L d_T(h, f) \leq \beta_L \varepsilon_{\mathrm{imp}}$ and similarly $|\hat{L}(S_n)[h] - \hat{L}(S_n)[f]| \leq \beta_{\hat{L}(S_n)} d_T(h, f) \leq \beta_{\hat{L}(S_n)} \varepsilon_{\mathrm{imp}}$.

**Conclusion: (Bias-Variance Decomposition)**  Then, the excess risk and generalization gap are bounded as follows: With probability at least $1 - \delta$ over an i.i.d. draw of sample $S_n \sim P$,

$$L[\hat{h}] - L_{\mathcal{C}} \leq \beta_L \varepsilon_{\text{imp}} + \varepsilon_{\text{model}} + 4\beta_\ell \hat{\mathfrak{R}}_{S_n}(\mathcal{H}_{\text{abs}}) + 2b\sqrt{\frac{2\log 1/\delta}{n}} \tag{7}$$

$$L[\hat{h}] - \hat{L}(S_n)[\hat{h}] \leq (\beta_L + \beta_{\hat{L}(S_n)})\varepsilon_{\text{imp}} + 2\beta_\ell \hat{\mathfrak{R}}_{S_n}(\mathcal{H}_{\text{abs}}) + b\sqrt{\frac{2\log 1/\delta}{n}}. \tag{8}$$

**Example of sufficient conditions:**  For example, the following setting satisfy the above assumptions:

- $\mathcal{Y} := \mathbb{R}$
- $\ell$ is 1-Lipschitz in the first argument a.e., i.e. $|\ell(y, y_0) - \ell(y', y_0)| \leq |y - y'|$ a.e. $y_0 \in \mathcal{Y}$
- $T(\mathcal{X}) := L^\infty(\mathcal{X})$, $\mathcal{C} := T(\mathcal{X})$, and $\mathcal{H}_{\text{abs}}, \mathcal{H}_{\text{imp}}$ are a separable and compact subspaces in $L^\infty(\mathcal{X})$

so that $|L[h] - L[f]| \leq \mathbb{E}_X|h(X) - f(X)| \leq \|h - f\|_\infty$ and $|\hat{L}(S_n)[h] - \hat{L}(S_n)[f]| \leq \frac{1}{n}\sum_{i=1}^n |h(X_i) - f(X_i)| \leq \|h - f\|_\infty$ for every $f, h \in L^\infty(\mathcal{X})$, $\beta_L = \beta_{\hat{L}(S_n)} = 1$, $L_{\mathcal{C}} = \inf_{c \in \mathcal{C}} L[c] = 0$, $\varepsilon_{\text{model}} = L_{\mathcal{H}_{\text{abs}}} - L_{\mathcal{C}} = \inf_{h \in \mathcal{H}_{\text{abs}}} L[h]$, and $\varepsilon_{\text{imp}} = \sup_{f \in \mathcal{H}_{\text{abs}}} \inf_{h \in \mathcal{H}_{\text{imp}}} \|h - f\|_\infty$.

*Proof.*  Write $\hat{h} := A(S_n) = \text{embed}(\hat{f})$.

$$L[\hat{h}] - L_{\mathcal{C}} = \underbrace{L[\hat{h}] - L[\hat{f}]}_{\text{implementation error} \leq \varepsilon_{\text{imp}}} + \underbrace{L[\hat{f}] - L_{\mathcal{C}}}_{\text{(i) excess risk wrt } \mathcal{H}_{\text{abs}}} \tag{9}$$

$$\text{(i) } L[\hat{f}] - L_{\mathcal{C}} = \underbrace{L[\hat{f}] - L_{\mathcal{H}_{\text{abs}}}}_{\text{(ii) variance}} + \underbrace{L_{\mathcal{H}_{\text{abs}}} - L_{\mathcal{C}}}_{\text{model bias } \varepsilon_{\text{model}}} \tag{10}$$

$$\text{(ii) } L[\hat{f}] - L_{\mathcal{H}_{\text{abs}}} = \underbrace{L[\hat{f}] - \hat{L}(S_n)[\hat{f}]}_{\text{generalization gap wrt } \mathcal{H}_{\text{abs}}} + \underbrace{\hat{L}(S_n)[\hat{f}] - \hat{L}(S_n)[f^\star]}_{\leq 0 \text{ by the minimality}} + \underbrace{\hat{L}(S_n)[f^\star] - L[f^\star]}_{\text{(estimation error)}} \tag{11}$$

$$\leq 2 \sup_{f \in \mathcal{H}_{\text{abs}}} |\hat{L}(S_n)[f] - L[f]| \tag{12}$$

$$\leq 4\beta_\ell \hat{\mathfrak{R}}_{S_n}(\mathcal{H}_{\text{abs}}) + 2b\sqrt{\frac{2\log 1/\delta}{n}} \quad \text{with prob.} \geq 1 - \delta \text{ by Theorem 6} \tag{13}$$

So, we have the following high probability bound:

$$L[\hat{h}] - L_{\mathcal{C}} \leq \beta_L \varepsilon_{\text{imp}} + \varepsilon_{\text{model}} + 4\hat{\mathfrak{R}}_{S_n}(\mathcal{H}_{\text{abs}}) + 2b\sqrt{2\frac{\log 1/\delta}{n}}. \tag{14}$$

Similarly, the generalization gap (= test error $-$ training error) is decomposed as follows:

$$L[\hat{h}] - \hat{L}(S_n)[\hat{h}] = \underbrace{L[\hat{h}] - L[\hat{f}]}_{\text{implementation error}} + \underbrace{L[\hat{f}] - \hat{L}(S_n)[\hat{f}]}_{\text{generalization gap wrt } \mathcal{H}_{\text{abs}}} + \underbrace{\hat{L}(S_n)[\hat{f}] - \hat{L}(S_n)[\hat{h}]}_{\text{implementation error}}$$

$$\leq \beta_L \varepsilon_{\text{imp}} + \sup_{f \in \mathcal{H}_{\text{abs}}} |L[f] - \hat{L}(S_n)[f]| + \beta_{\hat{L}(S_n)} \varepsilon_{\text{imp}}$$

$$\leq (\beta_L + \beta_{\hat{L}(S_n)})\varepsilon_{\text{imp}} + 2\beta_\ell \hat{\mathfrak{R}}_{S_n}(\mathcal{H}_{\text{abs}}) + b\sqrt{\frac{2\log 1/\delta}{n}} \quad \text{w. p.} \geq 1 - \delta \tag{15}$$

$$\square$$

## C  PROOF OF THEOREM 2

Let $(\mathcal{X}, d)$ be a pseudo-metric space, let $C(\mathcal{X})$ be the normed space of real-valued continuous functions $h : \mathcal{X} \to \mathbb{R}$ equipped with uniform norm $\| \bullet \|_\infty$, and let $C(\mathcal{X}, \mathcal{X})$ be the pseudo-metric space of continuous maps $f : \mathcal{X} \to \mathcal{X}$ equipped with the induced uniform pseudo-metric $d_\infty(f, f') := \sup_{x \in \mathcal{X}} d(f(x), f'(x))$. Additionally, given a finite sample $S = \{X_1, \ldots, X_n\} \subset \mathcal{X}$, let $d_S$ denote an induced empirical pseudo-metric $d_S(f, f') := \left(\frac{1}{n}\sum_{x \in S} d(f(x), f'(x))^2\right)^{1/2}$.

**Theorem 3** (Theorem 2, restated). *For any L-Lipschitz class $H \subset C(\mathcal{X})$ and any totally-bounded continuous class $F \subset C(\mathcal{X}, \mathcal{X})$ with identity element $\mathrm{id}_{\mathcal{X}}$ equipped with pseudo-metric $d_S$, the Rademacher complexity of composition class $H \circ F := \{h \circ f : \mathcal{X} \to \mathbb{R} \mid h \in H, f \in F\}$ is decomposed into the sum of the complexites of output class $H$ and hidden class $F$ as follows:*

$$\hat{\mathfrak{R}}_S(H \circ F) \leq \hat{\mathfrak{R}}_S(H) + \frac{12L}{\sqrt{n}} \int_0^{\mathrm{diam}(F)} \sqrt{\log N(F, d_S, \varepsilon)} d\varepsilon, \tag{16}$$

$$\mathfrak{R}_n(H \circ F) \leq \mathfrak{R}_n(H) + \frac{12L}{\sqrt{n}} \mathbb{E}_S \left[ \int_0^{\mathrm{diam}(F)} \sqrt{\log N(F, d_S, \varepsilon)} d\varepsilon \right]. \tag{17}$$

*Here $\mathfrak{R}_n(H)$ denotes the Rademacher complexity of $H$, $\log N(F, d_\infty, \varepsilon)$ denotes the metric entropy of $F$ in pseudo-metric $d_\infty$, and $\mathrm{diam}(F) := \sup_{f \in F} d_S(f, \mathrm{id}_{\mathcal{X}})/2$.*

*Proof.* For the fixed sample $S = \{X_1, \ldots, X_n\}$ and the Rademacher vector $\sigma = (\sigma_1, \ldots, \sigma_n) \in \{\pm 1\}^n$ define the process

$$Z_f(S, H, \sigma) := \frac{1}{n} \sup_{h \in H} \sum_{i=1}^n \sigma_i h(f(X_i)), \qquad f \in F. \tag{18}$$

Take an arbitrary anchor element $f_0 \in F$ and write the centered version

$$\widetilde{Z}_f := Z_f - Z_{f_0}, \quad \text{so that} \quad \widetilde{Z}_{f_0} = 0. \tag{19}$$

Because every $h \in H$ is $L$-Lipschitz w.r.t. $d$,

$$\left| h(f(x)) - h(f'(x)) \right| \leq Ld\left( f(x), f'(x) \right), \quad \forall x \in \mathcal{X} \tag{20}$$

hence for the difference of the processes

$$\begin{aligned}
\left| \widetilde{Z}_f - \widetilde{Z}_{f'} \right| &= \left| Z_f - Z_{f'} \right| \\
&= \left| \frac{1}{n} \sup_{h \in H} \sum_{i=1}^n \sigma_i h(f(X_i)) - \frac{1}{n} \sup_{h \in H} \sum_{i=1}^n \sigma_i h(f'(X_i)) \right| \\
&\leq \frac{1}{n} \sup_{h \in H} \sum_{i=1}^n \left| h(f(X_i)) - h(f'(X_i)) \right| \\
&\leq \frac{L}{n} \sum_{i=1}^n d(f(X_i), f'(X_i)) \leq Ld_S(f, f').
\end{aligned} \tag{21}$$

Conditioned on the sample $S$, the random variable $\widetilde{Z}_f - \widetilde{Z}_{f'}$ is a Rademacher (hence sub-Gaussian) sum with variance proxy at most $L^2 d_S(f, f')^2$. Formally,

$$\mathbb{E}_\sigma \exp\left( \lambda \left[ \widetilde{Z}_f - \widetilde{Z}_{f'} \right] \right) \leq \exp\left( \frac{\lambda^2}{2} L^2 d_S(f, f')^2 \right), \qquad \forall \lambda \in \mathbb{R}, \tag{22}$$

so $(\widetilde{Z}_f)_{f \in F}$ is a centered sub-Gaussian process indexed by the pseudo-metric space $(F, d_S)$ with variance parameter $L$.

The Dudley's integral bound (Martin J. Wainwright, 2019, Theorem 5.22) states that for such a centered sub-Gaussian process

$$\mathbb{E}_\sigma \sup_{f \in F} \widetilde{Z}_f \leq \frac{CL}{\sqrt{n}} \int_0^{\mathrm{diam}(F)} \sqrt{\log N(F, d_S, u)} du, \tag{23}$$

where $C \leq 12$ is a universal constant. (The integral starts at 0 but the integrand is clipped at the bound of $\widetilde{Z}$, hence the upper limit $\sup_{f \in F} d_S(f, f_0) \leq \mathrm{diam}(F)$.)

Un-centering $\widetilde{Z}_f$, we get

$$\mathbb{E}_\sigma \sup_{f \in F} Z_f \le \mathbb{E}_\sigma \left[ \sup_{f \in F} [Z_f - Z_{f_0}] + \sup_{f \in F} Z_{f_0} \right]$$

$$\le \mathbb{E}_\sigma Z_{f_0} + \frac{12L}{\sqrt{n}} \int_0^{\mathrm{diam}(F)} \sqrt{\log N(F, d_S, u)} du, \tag{24}$$

Because the expectation is only over $\sigma$, Eq. (24) already gives the empirical Rademacher bound $\hat{\mathfrak{R}}_S(H \circ F)$ stated previously; taking the outer expectation over samples produces the population bound.

If the identity map lies in the class $F$, i.e.

$$\mathrm{id}_\mathcal{X} \in F, \tag{25}$$

then we can simply take $f_0 = \mathrm{id}_\mathcal{X}$. Two consequences follow: First, the leading term becomes the ordinary empirical complexity of $H$.

$$\mathbb{E}_\sigma Z_{f_0} = \frac{1}{n} \sup_{h \in H} \sum_{i=1}^n \sigma_i h(X_i) = \hat{\mathfrak{R}}_S(H). \tag{26}$$

No extra optimization or data-dependent choice of $f_0$ is needed. Second, the right-hand integral is unchanged. The second term still measures only how flexibly the maps in $F$ can transport points (through $N(F, d_S, u)$) and is totally independent of the special anchor we used. Hence, with $\mathrm{id}_\mathcal{X} \in F$ the empirical bound specializes to

$$\hat{\mathfrak{R}}_S(H \circ F) \le \hat{\mathfrak{R}}_S(H) + \frac{12L}{\sqrt{n}} \int_0^{\mathrm{diam}(F)} \sqrt{\log N(F, d_S, u)} du. \tag{27}$$

The algebra and all probabilistic ingredients are exactly the same; the anchor is just chosen once and for all, which simplifies both notation and interpretation. □

## D A SIMPLER VERSION OF THEOREM 2

We assume the composition structure $\mathcal{H}_{\mathrm{abs}} = H \circ F$ for the abstract model class, where $H$ and $F$ respectively are collections of real-valued continuous functions $h_o : \mathcal{X} \to \mathbb{R}$ and continuous self-maps $f : \mathcal{X} \to \mathcal{X}$, i.e. $H \subset C(\mathcal{X})$ and $F \subset C(\mathcal{X}, \mathcal{X})$. In the following, we assume all $\mathcal{X}, C(\mathcal{X}), C(\mathcal{X}, \mathcal{X})$ are (pseudo)metric spaces to use covering numbers, every $h \in H$ is $B$-bounded and $L$-Lipschitz, and $F$ is equicontinuous and its *range* $F(\mathcal{X})$ is compact, so that $F$ and $H$ are totally bounded thus their covering numbers are finite.

Let $S = (X_1, \ldots, X_n)$ be a sample from $\mathcal{X}$ and define the empirical $L_2(S)$ metric $\|h - h'\|_S := \sqrt{\frac{1}{n} \sum_{i=1}^n \big(h(X_i) - h'(X_i)\big)^2}$. According to *Dudley's entropy integral* (Theorem 8), the Rademacher complexity $\hat{\mathfrak{R}}_S(H \circ F)$ is bounded by the logarithm of the covering number (*metric entropy*) of hypothesis class. Since $\|h - h'\|_S \le \|h - h'\|_\infty$, the covering number in $L_2(S)$ is no larger than that in $\|\cdot\|_\infty$. So Dudley's bound yields

$$\hat{\mathfrak{R}}_S(H \circ F) \le \frac{12}{\sqrt{n}} \int_0^{\mathrm{diam}(H \circ F, \|\cdot\|_S)} \sqrt{\log N\big(H \circ F, \|\cdot\|_S, \varepsilon\big)} d\varepsilon \tag{28}$$

$$\le \frac{12}{\sqrt{n}} \int_0^B \sqrt{\log N\big(H \circ F, \|\cdot\|_\infty, \varepsilon\big)} d\varepsilon. \tag{29}$$

Then, according to Lemma 8, the covering number in the uniform norm of the composite class $H \circ F$ can be decomposed into a product of covering numbers; or in terms of the Rademacher complexity bound, the sum of metric entropies: For every positive number $\varepsilon > 0$,

$$\log N(H \circ F, \|\cdot\|_\infty, \varepsilon) \le \log N(H, \|\cdot\|_\infty, \varepsilon/2) + \log N(F, d_\infty, \varepsilon/(2L)). \tag{30}$$

Insert Eq. (30) and use $\sqrt{a+b} \le \sqrt{a} + \sqrt{b}$, we have the following decomposition of the Rademacher complexity for composition class:

**Theorem 4** (Theorem 2, simpler parallel). *For any sample $S$,*

$$\hat{\mathfrak{R}}_S(H \circ F) \leq \frac{12}{\sqrt{n}} \int_0^B \left\{ \sqrt{\log N(H, \|\cdot\|_\infty, \varepsilon/2)} + \sqrt{\log N(F, d_\infty, \varepsilon/(2L))} \right\} \mathrm{d}\varepsilon. \tag{31}$$

*Averaging over samples gives the same bound for $\mathfrak{R}_n(H \circ F)$. Changing variables ($u = \varepsilon/2, v = \varepsilon/(2L)$), yields a completely explicit decomposition:*

$$\mathfrak{R}_n(H \circ F) \leq \frac{24}{\sqrt{n}} \Big[ \int_0^{B/2} \sqrt{\log N(H, \|\cdot\|_\infty, u)} \mathrm{d}u + L \int_0^{B/(2L)} \sqrt{\log N(F, d_\infty, v)} \mathrm{d}v \Big]. \tag{32}$$

These inequalities decompose the statistical complexity of the composite class into separate contributions from the feature layer $F$ and the output layer $H$.

# E   DETAILS ON GROWTH RATE ANALYSIS

We provide details on the conditions and examples overviewed in Section 4.

## E.1   ASSUMPTIONS AND NOTATION

Throughout, $(\mathcal{X}, d)$ is a (not necessarily compact) metric space, and $C(\mathcal{X}, \mathcal{X})$ denotes the set of continuous self-maps of $\mathcal{X}$.

**Uniform metric**   On $C(\mathcal{X}, \mathcal{X})$ we use the uniform metric

$$d_\infty(f, g) := \sup_{x \in \mathcal{X}} d\left( f(x), g(x) \right).$$

When $\mathcal{X}$ is non-compact, $d_\infty$ is not well-defined as the metric may diverge. We will always guarantee finiteness in one of the following two ways (either assumption suffices):

- ($d_b$) replace $d$ by its bounded version $d_b(x, y) := \min\{1, d(x, y)\}$;
- ($C_b$) fix a compact set $K_0 \subset \mathcal{X}$ and assume all maps under discussion (and all words built from them) take values in $K_0$.

All the three metrics—$C(\mathcal{X}, \mathcal{X})$ with compact $\mathcal{X}$ and $d_\infty$, $C(\mathcal{X}, \mathcal{X})$ with non-compact $\mathcal{X}$ and $d_b$, and $C_b(\mathcal{X}, \mathcal{X})$ with non-compact $\mathcal{X}$ and $d_\infty$—induce the uniform topology.

**Arzelà–Ascoli and compact-open topology**   Arzelà–Ascoli theorems assert relative compactness in the *compact-open topology (COT)*, which is *weaker* than uniform topology (UT) in general. COT is equivalent to the compact-convergence topology, or the topology induced from the uniform convergence on every compact sets. COT *coincides with* UT when $\mathcal{X}$ is compact. See Appendix I for more details on Arzelà–Ascoli theorems.

**Homeo and isometry groups**   $\mathrm{Homeo}(\mathcal{X}) \subset C(\mathcal{X}, \mathcal{X})$ denotes the homeomorphism group of $\mathcal{X}$, that is, the set of all *bijective* bi-continuous self-maps with the function composition as group operation. $\mathrm{Isom}(\mathcal{X}) \subset C(\mathcal{X}, \mathcal{X})$ denotes the isometry group of $(\mathcal{X}, d)$, that is, the set of all *bijective* distance-perserving self-maps with the function composition as group operation. The above group operations are continuous in compact-open topology, or the topology induced by the following metric $d_\infty$ (restricted to an arbitrary compact subset $K \subset \mathcal{X}$). Hence $\mathrm{Homeo}(\mathcal{X})$ and $\mathrm{Iso}(\mathcal{X})$ are topological groups.

**Semigroup and word balls**   For $F \subset C(\mathcal{X}, \mathcal{X})$, write $\langle F \rangle$ for the semigroup generated by $F$ under composition. For $k \in \mathbb{N}$, let $B(k, F)$ be the set of maps obtainable as a composition of at most $k$ maps from $F$.

**Covering and packing numbers**  For a metric space $(\mathcal{M}, \rho)$, write

$$N(A, \rho, \varepsilon) := \min \left\{ |C| \; \middle| \; C \subset \mathcal{M} \text{ s.t. } A \subset \bigcup_{y \in C} B_\rho(y, \varepsilon) \right\}$$

for the $\varepsilon$-covering number of $A \subset \mathcal{M}$, and

$$M(A, \rho, \varepsilon) := \max \left\{ |S| \mid S \subset A, \rho(x, y) \geq \varepsilon \; \forall x \neq y \in S \right\}$$

for the $\varepsilon$-packing number. See also Appendix H for more details on covering and packing numbers.

**Lipschitz and co-Lipschitz constants**  For $f \in C(\mathcal{X}, \mathcal{X})$ and $S \subset \mathcal{X}$ nonempty,

$$\mathrm{Lip}_S(f) := \sup_{x \neq y \in S} \frac{d(f(x), f(y))}{d(x, y)}, \qquad \mathrm{coLip}_S(f) := \inf_{x \neq y \in S} \frac{d(f(x), f(y))}{d(x, y)}.$$

We say "$f$ is uniformly expanding on $S$" if $\mathrm{coLip}_S(f) > 1$.

**Word length in $\langle F \rangle$.**  For $h \in \langle F \rangle$, denote by $\ell_F(h)$ the least $\ell$ such that $h$ is a composition of $\ell$ maps from $F$. In all results below, when we assert that certain auxiliary maps $h$ belong to $\langle F \rangle$, we also assume that $\ell_F(h)$ is bounded by a constant independent of $k$.

**Subshift with ultrametric**  On the *one-sided full shift* $\Sigma_m^+ = [m]^{\mathbb{N}}$ (with $[m] = \{1, \ldots, m\}$ discrete), fix $\theta \in (0, 1)$ and define the ultrametric

$$d_\theta(x, y) = \begin{cases} 0, & x = y, \\ \theta^{n(x,y)-1}, & x \neq y, \end{cases} \quad \text{where} \quad n(x, y) := \min\{i \geq 1 \mid x_i \neq y_i\}.$$

This is an ultrametric (satisfies *strong triangle inequality* $d(x, z) \leq \max\{d(x, y), d(y, z)\}$), it induces the product topology, and it makes $\Sigma_m^+$ a compact, totally disconnected, perfect (Cantor-type) space. Distances take only the values $1, \theta, \theta^2, \ldots$; in particular, sequences that differ in the first symbol are at distance $1$.

For a finite word $u = u_1 \cdots u_\ell$, the *(prefix) cylinder* of depth $\ell$ is given by

$$\llbracket u \rrbracket := \{w \in \Sigma_m^+ \mid w_1 = u_1, \ldots, w_\ell = u_\ell\},$$

a clopen set; cylinders form a basis and the depth-$\ell$ cylinders partition $\Sigma_m^+$ into $m^\ell$ pieces. In the metric $d_\theta$, $\mathrm{diam}(\llbracket u \rrbracket) = \theta^\ell$, and any two distinct depth-$\ell$ cylinders are separated by at least $\theta^{\ell-1}$. Moreover, balls coincide with cylinders: for $w \in \Sigma_m^+$ and $\ell \geq 0$, $\overline{B}_{d_\theta}(w, \theta^\ell) = \llbracket w_1 \cdots w_\ell \rrbracket$. The left shift $\sigma(w)_i = w_{i+1}$ satisfies $\sigma(\llbracket au \rrbracket) = \llbracket u \rrbracket$ and $\sigma^{-1}(\llbracket u \rrbracket) = \bigsqcup_{a \in [m]} \llbracket au \rrbracket$.

### E.2  BASIC FACTS ON COVERING AND PACKING NUMBERS

**Lemma 1** (Packing-Covering). *For any totally bounded metric space $A$ and $\varepsilon > 0$,*

$$M(A, 2\varepsilon) \leq N(A, \varepsilon) \leq M(A, \varepsilon).$$

**Lemma 2** (Lipschitz Embedding). *Let $(X, d_X), (Y, d_Y)$ be metric spaces. Suppose $\phi : X \to Y$ is $L$-Lipschitz, then for every subset $S \subset X$,*

$$N(\phi(S), d_Y, L\varepsilon) \leq N(S, d_X, \varepsilon), \quad M(\phi(S), d_Y, L\varepsilon) \leq M(S, d_X, \varepsilon).$$

**Lemma 3** (Subadditivity). *For any metric space $(M, d)$, subsets $F, G \subset M$, and $\varepsilon > 0$, we have*

$$N(F \cup G, \varepsilon) \leq N(F, \varepsilon) + N(G, \varepsilon).$$

*Proof.* Let $A, B$ be $\varepsilon$-coverings of $F, G$ respectively. Then, $A \cup B$ is an $\varepsilon$-covering of $F \cup G$ because for any $h \in F \cup G$ there exists $c \in A \cup B$ satisfying $d(h, c) \leq \varepsilon$. Thus, $N(F \cup G, \varepsilon) \leq |A \cup B| \leq |A| + |B| = N(F, \varepsilon) + N(G, \varepsilon)$. $\qquad\qquad\square$

**Lemma 4** (Sub-multiplicativity). *For any bi-Lipschitz metric semigroup $(M, d)$ with $\sup_{f \in M} d(fx, fy) \leq \lambda d(x, y)$ (left Lipschitz) and $\sup_{f \in M} d(xf, yf) \leq \rho d(x, y)$ (right Lipschitz), for any subsets $F, G \subset M$, and $\varepsilon, \delta > 0$, we have*

$$N(FG, \varepsilon + \delta) \leq N(F, \varepsilon/\rho) N(G, \delta/\lambda).$$

*Proof.* Let $A, B$ be $\alpha, \beta$-coverings of $F, G$ respectively. Then, $AB$ is an $(\rho\alpha + \lambda\beta)$-covering of $FG$ because for any $fg \in FG$ there exists $ab \in AB$ satisfying $d(fg, ab) \leq d(fg, ag) + d(ag, ab) \leq \rho d(f, a) + \lambda d(g, b) = \rho\alpha + \lambda\beta$. Thus, $N(FG, \rho\alpha + \lambda\beta) \leq |AB| \leq |A||B| = N(F, \alpha)N(G, \beta)$. Letting $\alpha = \varepsilon/\rho, \beta = \delta/\lambda$ yields the assertion. $\qquad\square$

Also recall:

**Lemma 5** (Right-Composition is 1-Lipschitz)**.** $d_\infty(a \circ f, b \circ f) \leq d_\infty(a, b)$

*Proof.* $d_\infty(a \circ f, b \circ f) = \sup_{x \in \mathcal{X}} d(a(f(x)), b(f(x))) \leq \sup_{y \in \mathcal{X}} d(a(y), b(y)) = d_\infty(a, b)$ $\qquad\square$

**Lemma 6** (Left-Composition is Lipschitz)**.** $d_\infty(f \circ a, f \circ b) \leq \mathrm{Lip}(f) d_\infty(a, b)$

*Proof.* $d_\infty(f \circ a, f \circ b) = \sup_{x \in \mathcal{X}} d(f(a(x)), f(b(x))) \leq \sup_{x \in \mathcal{X}} \mathrm{Lip}(f) d(a(x), b(x)) = \mathrm{Lip}(f) d_\infty(a, b)$ $\qquad\square$

**Lemma 7** (Packing–Covering via Finite Probes)**.** *Let* $P = \{x_1, \ldots, x_m\} \subset \mathcal{X}$ *be finite and define* $d_P((y_j), (z_j)) := \max_{1 \leq j \leq m} d(y_j, z_j)$ *on* $\mathcal{X}^m$, *and* $\mathrm{ev}_P : C(\mathcal{X}, \mathcal{X}) \to \mathcal{X}^m, \mathrm{ev}_P(f) = (f(x_1), \ldots, f(x_m))$. *Then* $\mathrm{ev}_P$ *is 1-Lipschitz:* $d_P(\mathrm{ev}_P(f), \mathrm{ev}_P(g)) \leq d_\infty(f, g)$. *If* $\mathrm{ev}_P(S) \subset \mathcal{X}^m$ *contains* $M$ *points that are pairwise* $\delta$*-separated (in* $d_P$*), then for every* $\varepsilon < \delta/2$,

$$N(S, d_\infty, \varepsilon) \geq M.$$

*Proof.* The 1-Lipschitz claim is immediate:

$$d(f(x_j), g(x_j)) \leq \sup_{x \in \mathcal{X}} d(f(x), g(x)) = d_\infty(f, g).$$

If $\mathrm{ev}_P(s_1), \ldots, \mathrm{ev}_P(s_M)$ are $\delta$-separated, then $d_\infty(s_i, s_j) \geq d_P(\mathrm{ev}_P(s_i), \mathrm{ev}_P(s_j)) \geq \delta$. Any ball of radius $\varepsilon < \delta/2$ in the $d_\infty$-metric can contain at most one of the $s_i$'s, so at least $M$ balls are needed. $\qquad\square$

### E.3 Conditions for Saturation and Polynomial Growth

#### E.3.1 P1. Equicontinuous Semigroup on Compact Domain $\implies$ Saturation in $k$

**Condition P1** (Equicontinuous Semigroup on Compact Domain Saturates)**.** *Assume* $\mathcal{X}$ *is compact. Suppose (at least) one of the following assumptions is satisfied:*

1. *Semigroup* $\langle F \rangle$ *is (pre)compact,*

2a. *Semigroup* $\langle F \rangle$ *is equicontinuous,*

2b. *Semigroup* $\langle F \rangle$ *is uniformly Lipschitz:* $\mathrm{Lip}\langle F \rangle < \infty$, *or*

2c. *Generators* $F$ *are non-expanding:* $\mathrm{Lip}\, F \leq 1$.

*Then, closure semigroup* $G := \overline{\langle F \rangle}^{d_\infty} \subset C(\mathcal{X}, \mathcal{X})$ *is compact and equicontinuous, and for all* $\varepsilon > 0$ *and all* $k$

$$N(B(k, F), \varepsilon) \leq N(G, \varepsilon) \quad (< \infty),$$

*hence no dependence on* $k$.

*Proof.* If $\mathcal{X}$ is compact, then $C(\mathcal{X}, \mathcal{X})$ is complete in $d_\infty$. So the closure of any precompact subset in $C(\mathcal{X}, \mathcal{X})$ is compact. Besides, Arzelà–Ascoli yields if $\mathcal{X}$ is compact, then (1) any compact subset of $C(\mathcal{X}, \mathcal{X})$ is equicontinuous, and (2) any equicontinuous subset is relatively compact (thus compact). (Namely, any subset in $C(\mathcal{X}, \mathcal{X})$ with compact $\mathcal{X}$ is precompact $\iff$ compact $\iff$ equicontinuous.) Both Assumptions 1 and 2a are straightforward. Since uniform Lipschitz implies equicontinuous, Assumptions 2b and 2c are reduced to Assumption 2a. Hence each assumption implies $G$ is compact (and equicontinuous). Trivial inclusion $B(k, F) \subseteq G$ yields the bound, and compactness of $G$ gives finiteness of $N(G, \varepsilon)$. $\qquad\square$

*Example* 1 (Rotations on Circle). Let $\mathcal{X} = \mathbb{S}^1$, $A \subset \mathbb{S}^1$ compact, and put $F := \{R_\alpha \mid \alpha \in A\}$ (rotations). All are isometries, and the closure of the generated subgroup $\langle F \rangle$ is a compact torus (either a finite set or the full circle group depending on rational relations) contained in the rotation group $G = O(1)$, so

$$N(B(k, F), d_\infty, \varepsilon) \le N(O(1), d_\infty, \varepsilon) = N(\mathbb{S}^1, d, \varepsilon) \quad (k\text{-independent}).$$

*Example* 2 (Finite Isometry Group on Finite Set). If $\mathcal{X}$ is finite and $F \subset \mathrm{Iso}(\mathcal{X})$, then $G$ is finite; $N(B(k, F), \varepsilon)$ is bounded by $|G|$ for all $k$.

**Condition P1'** (Contraction to Compact Invariant Set). *Assume*

1. *(uniform contraction)* $\sup_{f \in F} \mathrm{Lip}(f) \le c < 1$,

2. *(compact attractor) there exists a nonempty compact $F$-invariant set $A \subset \mathcal{X}$ (i.e., $f(A) \subset A$ for all $f \in F$), and*

3. *(compact absorbing set) there exist $L \in \mathbb{N}$ and bounded set $K \subset \mathcal{X}$ such that $f(\mathcal{X}) \subset K$ for all depth-$L$ map $f \in F^L$.*

*Then for every $\varepsilon > 0$ there exists*

$$m(\varepsilon) := L + \left\lceil \log_{1/c}\left(\frac{2\,\mathrm{diam}(K)}{\varepsilon}\right) \right\rceil$$

*such that for all $k \ge m(\varepsilon)$,*

$$N\left(B(k, F), d_\infty, \varepsilon\right) \le N\left(A, d, \tfrac{\varepsilon}{2}\right) + \sum_{\ell=0}^{m(\varepsilon)-1} \left[N\left(F, d_\infty, \tfrac{\varepsilon}{2}\right)\right]^\ell.$$

*In particular, the right-hand side is* independent *of $k$, so growth in $k$ saturates at any fixed $\varepsilon$ to the entropy of the attractor $A$.*

*Remark* 2. When $\mathcal{X}$ is compact, we can simply set $A = K = \mathcal{X}$ and $L = 0$.

*Remark* 3. A single saturation map $\sigma$ such as $\tanh$, logistic map, and clipping yields bounded absorbing set $K = \sigma(\mathcal{X})$ with $L = 1$.

*Proof.* Pick any $a_0 \in A$. For any word $w \in F^\ell$, we have $\mathrm{Lip}(w) \le c^\ell$ and $w(a_0) \in A$ (invariance). If $\ell \ge m(\varepsilon)$, then

$$\sup_{x \in \mathcal{X}} d\left(w(x), w(a_0)\right) \le c^{\ell - L}\,\mathrm{diam}(K), \quad \text{(bounded absorbing set)}$$

and the right-hand side is $\le \varepsilon/2$, i.e.

$$d_\infty\left(w, \mathrm{const}_{w(a_0)}\right) \le \varepsilon/2.$$

Hence all sufficiently deep words, i.e. $\bigcup_{\ell \ge m(\varepsilon)} F^\ell$, are covered by the $\varepsilon/2$-thickening of the set of constant maps landing in $A$: $\mathrm{const}_q : q \in A$. These constants at resolution $\varepsilon/2$ are parameterized by an $\varepsilon/2$-net of $A$, giving the bound $N(A, \varepsilon/2)$. For the finitely many shallow layers $\ell < m(\varepsilon)$, use the submultiplicative covering inequality for composition to get $N(F^\ell, \varepsilon) \le [N(F, \varepsilon/2)]^\ell$, and sum over $\ell$. $\qquad \square$

Intuitively, deep words are *almost constant* (their images have diameter $\le c^{\ell - L}\,\mathrm{diam}(K)$), so at scale $\varepsilon$ only the landing point $w(a_0) \in A$ matters; hence the dependence on $k$ disappears once $\ell$ exceeds an $\varepsilon$-dependent "memory length" $m(\varepsilon)$.

*Example* 3 (Cantor Attractor (Compact, Free, Contraction)). Let $\mathcal{X} = [0, 1]$ and $F = f_0(x) = x/3$, $f_1(x) = (x + 2)/3$. Then $\sup \mathrm{Lip}(f_i) = 1/3 = c < 1$. The attractor $A$ is the middle–third Cantor set $C$ (compact, $F$-invariant). By Condition P1', for every $\varepsilon > 0$ and $k \ge m(\varepsilon) = \lceil \log_3(2/\varepsilon) \rceil$,

$$N\left(B(k, F), \varepsilon\right) \lesssim N\left(C, \varepsilon/2\right),$$

independent of $k$. (Indeed, every $w \in F^\ell$ has the form $w(x) = 3^{-\ell} x + b_w$ and hence is $\varepsilon/2$-close to the constant map $x \mapsto b_w \in C$.)

**Condition P2** (Bi-Lipschitz Nilpotent Lie Group Action Grows Polynomially). *Let $H$ be a connected, simply-connected nilpotent Lie group, let $d_H$ be a left-invariant Riemannian metric on $H$, and let $D$ be the Guivarc'h–Bass homogeneous dimension of $H$. Assume that there is a faithful action $\alpha : H \to \mathrm{Homeo}(\mathcal{X})$ such that $\langle F \rangle \subset \alpha(H)$, and suppose bi-Lipschitz control: There exist constants $0 < c \leq C < \infty$ such that for all $g, h \in H$,*

$$cd_H(g,h) \leq d_\infty\left(\alpha(g), \alpha(h)\right) \leq Cd_H(g,h).$$

*Then, there exist a constant $C$ such that for every $\varepsilon > 0$ and $k \geq 1$,*

$$N\left(B(k,F), \varepsilon\right) \leq C\left(1 + \frac{k}{\varepsilon}\right)^D.$$

*Remark* 4. The Guivarc'h–Bass homogeneous dimension bridges the algebraic property such as the nilpotency of Lie group $H$ with the geometric property, namely the polynomial volume growth rate of $H$. We briefly reviewed the theory in Appendix J based on Breuillard (2014).

*Proof.* Let $S := \alpha^{-1}(F) \subset H$, a compact generating set for the subgroup $H_0 := \langle S \rangle$ (closed in $H$). Let $\ell_S$ be the word length with respect to $S$, let $B[\ell_S](k) := \{h \in H \mid \ell_S(h) \leq k\}$ be the word ball of length at most $k$, and let $B[d_H](k) := \{h \in H \mid d_H(e,h) \leq k\}$ be the geodesic ball of radius $k$. Any word of length $\leq k$ in $F$ equals $\alpha(g)$ with $g \in B[\ell_S](k)$. By standard comparability of word and geodesic metrics on nilpotent Lie groups, there is $A \geq 1$ so that

$$B[\ell_S](k) = \{h \in H \mid \ell_S(h) \leq k\} \quad \subset \quad B[d_H](Ak) = \{h \in H \mid d_H(e,h) \leq Ak\}.$$

By the bi-Lipschitz control,

$$N\left(\alpha(B[d_H](Ak)), d_\infty, \varepsilon\right) \leq N\left(B[d_H](Ak), d_H, \varepsilon/C\right).$$

Balls in $H$ have polynomial metric entropy: since $H$ has homogeneous dimension $D$, one has $N(B[d_H](R), \delta) \leq C'(1 + \frac{R}{\delta})^D$ for every $R >$ and $\delta > 0$ (cover by $\delta$-lattices in exponential coordinates). Therefore

$$N\left(B(k,F), d_\infty, \varepsilon\right) \leq N\left(\alpha(B[d_H](Ak)), d_\infty, \varepsilon\right) \leq C''\left(1 + k/\varepsilon\right)^D.$$

$$\square$$

*Example* 4 (Translations on Euclidean space (Non-compact, Abelian, Isometric)). Let $\mathcal{X} = \mathbb{R}^d$, $H = \mathbb{R}^d$ acting by translation $\alpha(v) := x \mapsto x + v, v \in H$. For any compact $A \subset H$, put $F := \alpha(A) \subset C(\mathcal{X}, \mathcal{X})$. The homogeneous dimension is $D = \dim \mathrm{span}\, A (\leq d)$; and $d_\infty(\alpha(v), \alpha(w)) = \|v - w\|$ (independent of $x$), so the bi-Lipschitz constants are $c = C = 1$. Therefore,

$$N(B(k,F), d_\infty, \varepsilon) = N(B(k,A), \|\cdot\|, \varepsilon) \lesssim (1 + k/\varepsilon)^D.$$

We note that if $\mathcal{X} = \mathbb{T}^d = \mathbb{R}^d/\mathbb{Z}^d$ (torus, compact) instead of $\mathbb{R}^d$ (non-compact), then it grows at $O((k/\varepsilon)^D)$ for early $k$, and saturates to $O(1/\varepsilon^D)$ by Condition P1.

*Example* 5 (Shear on Torus (Compact, Abelian, Expanding)). Let $\mathcal{X} = \mathbb{T}^2$, $H = \mathbb{Z}$ acting by shear $\alpha(n) := (x,y) \mapsto (x, x + ny), n \in H$. Put $F := \alpha(1) \subset C(\mathcal{X}, \mathcal{X})$. The homogeneous dimension is $D = 1$; and $d_\infty(\alpha(n), \alpha(m)) = \delta_{nm}$, so the bi-Lipschitz constants are $c = C = 1$. Therefore for $\varepsilon < 1$,

$$N(B(k,F), d_\infty, \varepsilon) = N(\{0\} \cup [k], \delta_{\bullet\bullet}, \varepsilon) = 1 + k.$$

*Example* 6 (Discrete Heisenberg Group on Torus (Compact, Nilpotent, Expanding)). Take $H$ the Heisenberg group acting affine-linearly on $\mathbb{T}^3$ (or $\mathbb{R}^3/\mathbb{Z}^3$) by translations along orbits. The homogeneous dimension is $D = 4$; with a compact generating set $F \subset \alpha(H)$, one gets $N(B(k,F), \varepsilon) \lesssim (1 + k/\varepsilon)^4$.

*Example* 7 (Upper-Triangular Unipotent Group on Euclidean Space (Non-compact, Nilpotent, Expanding)). Let $\mathcal{X} = \mathbb{R}^d$, $H = UT_d(\mathbb{R}) = \{I_d + N_d \mid N_d \in GL_d(\mathbb{R})$ strictly uppder triangular$\}$ acting by a bi-Lipschitz $\alpha$. Suppose a compact generating set $F \subset \alpha(H)$. The homogeneous dimension of $H$ is $D = \frac{d(d-1)(d+1)}{6}$. So,

$$N(B(k,F), d_\infty, \varepsilon) \asymp (1 + k/\varepsilon)^D.$$

## E.4 Conditions for (Super-/Double-)Exponential Growths

### E.4.1 E1: Free Semigroup + One–Point Uniform Separation $\implies$ Exponential Lowerbounds in $k$

**Condition E1** (Free Semigroup + Point Uniform Separation). *Let $F = \{f_1, \ldots, f_r\} \subset C(\mathcal{X}, \mathcal{X})$ with $r \geq 2$. Assume:*

1. *(Freeness at each length) For every $k$, the map $u \mapsto f_u := f_{i_k} \circ \cdots \circ f_{i_1}$ is injective on words $u \in [r]^k$ (i.e., the semigroup generated by $F$ is free on these generators).*

2. *(Uniform separation at a base point) There exist $x_* \in \mathcal{X}$ and $\delta > 0$ such that for all $k$ and all distinct words $u, v$ of length $k$, $d\big(f_u(x_*), f_v(x_*)\big) \geq \delta$.*

*Then, for every $k$ and every $\varepsilon < \delta/2$,*

$$N\big(B(k, F), d_\infty, \varepsilon\big) \geq r^k.$$

*Proof.* Apply Lemma 7 with $P = \{x_*\}$. For each fixed $k$, the $r^k$ words of length $k$ give $r^k$ maps whose images at $x_*$ are $\delta$-separated; hence for $\varepsilon < \delta/2$ at least $r^k$ balls are needed to cover $B(k, F)$. $\square$

*Example* 8 (Free Group with Word Metric (Non-Compact, Free, Isometric)). Let $\mathcal{X} = F_r$ ($r \geq 2$) with the standard word metric $d_G(u, v) = |u^{-1}v|$ (reduced word length), and let $F = \{L_{a_1}, \ldots, L_{a_r}\}$ where $L_{a_i}(x) = a_i x$. Each $L_{a_i}$ is an isometry of $(\mathcal{X}, d_G)$; the semigroup is free. Take $x_* = e$ (identity). If $u \neq v$ are words of the same length $k$, then $u^{-1}v$ is a nontrivial reduced word of length $\geq 2$ (recall "first rightmost mismatch gives $a^{-1}b$"). Hence

$$d_G\big(L_u(x_*), L_v(x_*)\big) = d_G(u, v) = |u^{-1}v| \geq 2,$$

so Condition E1 applies with $\delta = 2$: for all $\varepsilon < 1$,

$$N\big(B(k, F), d_\infty, \varepsilon\big) \geq r^k.$$

*Remark* 5. For distinct $g, h \in F_r$,

$$d_\infty(L_g, L_h) = \sup_x d_G(gx, hx) = \sup_x d_G\big(e, x^{-1}g^{-1}hx\big) = \infty,$$

since $|x^{-1}g^{-1}hx| \to \infty$ along $x = s^n$ with $s$ avoiding the boundary letters of $g^{-1}h$. This does *not* harm the lower bound: we only need that $d_\infty(L_u, L_v) \geq d_G\big(L_u(e), L_v(e)\big) \geq 2$, then apply Lemma 7 with $\varepsilon < 1$.

**Condition E1'** (Isometry with Coarse Coding). *Assume $F \subset \mathrm{Iso}(\mathcal{X})$ and that $d_\infty(g, h) < \infty$ for all $g, h \in \langle F \rangle$ (this holds, for example, when $d$ is bi-invariant: $d(axb, ayb) = d(x, y)$). Suppose furthermore:*

1. *(Freeness) The semigroup generated by $F$ is free (no relations in positive words).*

2. *(Coarse embedding at a base point) There exist $x_* \in \mathcal{X}$ and $c > 0$ such that for all $u, v \in \langle F \rangle$,*

$$d\big(u(x_*), v(x_*)\big) \geq c \cdot \mathrm{dist}_{\mathrm{word}}(u, v),$$

   *where $\mathrm{dist}_{\mathrm{word}}$ is the usual combinatorial distance on the free semigroup.*

*Then, for every $k$ and every $\varepsilon < c/2$,*

$$N\big(B(k, F), d_\infty, \varepsilon\big) \geq r^k.$$

*Proof.* Since $u \mapsto u(x_*)$ is $c$-Lipschitz from below with respect to word distance, distinct words of the same length are $\geq c$-separated at $x_*$. Apply Lemma 7 with $P = \{x_*\}$. The finiteness of $d_\infty$ guarantees that covering numbers are meaningful. $\square$

*Example* 9 (Free Group with Bi-Invariant Metric (Non-compact, Free, Isometric)). Let $\mathcal{X} = G = F_r$ $(r \geq 2)$. Let

$$S := \left\{ x a_i x^{-1}, x a_i^{-1} x^{-1} \mid x \in F_r, i = 1, \dots, r \right\}$$

and define the *conjugacy-invariant* word metric

$$d_S(g, h) := |g^{-1} h|_S,$$

the shortest length in the alphabet $S$. This metric is *bi-invariant*, hence all left translations $L_g$ are isometries and, crucially,

$$d_\infty(L_g, L_h) = \sup_x d_S(gx, hx) = \sup_x d_S\big(e, x^{-1} g^{-1} hx\big) = d_S(e, g^{-1} h) < \infty$$

(the supremum is independent of $x$).

Let $F = \{L_{a_1}, \dots, L_{a_r}\}$. The positive semigroup is free. With $x_* = e$, for words $u \neq v$ of the same length,

$$d_S\big(L_u(x_*), L_v(x_*)\big) = d_S(u, v) = |u^{-1} v|_S \geq 1,$$

so Condition E1' applies with $c = 1$: for every $\varepsilon < 1/2$,

$$N\big(B(k, F), d_\infty, \varepsilon\big) \geq r^k.$$

Unlike Example 8, $d_\infty$ is **finite** for all pairs.

### E.4.2 E2: PING–PONG WITH NON-CONTRACTION $\implies$ EXPONENTIAL LOWERBOUNDS IN $k$

**Condition E2** (Ping–Pong). *Let $F = \{f_1, \dots, f_r\} \subset C(\mathcal{X}, \mathcal{X})$ with $r \geq 2$. Suppose there are pairwise separated non-empty sets (chambers) $U_1, \dots, U_r \subset \mathcal{X}$, points (anchors) $a_1, \dots, a_r \in \mathcal{X}$, and constants $\Delta, \delta_0 > 0$ such that:*

- *(PP1: Pairwise separation)* $\operatorname{dist}(U_i, U_j) \geq \Delta$ *for all $i \neq j$.*

- *(PP2: Expansion on own domain)* *For each $i$, $f_i$ maps $U_i$ onto a set that meets every $U_j$.*

- *(PP3: Reset off domain)* *On $\mathcal{X} \setminus U_i$, $f_i$ stays within $\delta_0$ of the constant $a_i$.*

- *(PP4: Anchors separated)* $\min_{i \neq j} d(a_i, a_j) > 0$.

*Then, there exists $\delta > 0$ (depending only on $\Delta, \delta_0, \{a_i\}$) such that for every $k$ and every $\varepsilon < \delta/2$,*

$$N\big(B(k, F), d_\infty, \varepsilon\big) \geq r^k.$$

*Moreover, the map $u \mapsto f_u$ is injective (freeness).*

*Proof.* For a word $u = i_k \cdots i_1$ construct $x_u \in U_{i_1}$ by backwards induction so that

$$f_{i_j}(x_{i_j \cdots i_1}) \in U_{i_{j+1}} \quad (1 \leq j \leq k-1),$$

which is possible by (PP2). Now fix distinct words $u = i_k \cdots i_1$ and $v = j_k \cdots j_1$ of the same length and let $t$ be the rightmost index with $i_t \neq j_t$. Then $f_{i_{t-1}} \cdots f_{i_1}(x_u) \in U_{i_t}$, but $f_{j_{t-1}} \cdots f_{j_1}(x_u) \notin U_{i_t}$. Applying $f_{i_t}$ vs. $f_{j_t}$ at that step, one output is close to $a_{j_t}$ (by reset), the other remains in a set well separated by $\Delta$. Hence

$$d\big(f_u(x_u), f_v(x_u)\big) \geq \delta$$

for some $\delta > 0$ determined by $\Delta, \delta_0, \{a_i\}$. Distinct words are therefore distinct maps (freeness), and Lemma 7 with $P = \{x_u \mid |u| = k\}$ (or even a single carefully chosen probe) gives the claimed lower bound, uniform in $k$. $\square$

*Remark* 6. The assumptions formalize the common "expand inside the chamber; collapse to an anchor outside" motif. Uniform expansion coLip $> 1$ is needed; only local expansion and a reset suffice.

*Example* 10 (Piecewise-Linear Ping-Pong (Compact, Expansive)). Let $\mathcal{X} = [0, 1]$ with Euclidean distance. Fix $\eta \in (0, 1/12)$ and set

$$I_0 = \left[0, \tfrac{1}{3} - \eta\right], \quad I_1 = \left[\tfrac{2}{3} + \eta, 1\right], \qquad J_0 = \left[\tfrac{1}{3} - \eta, \tfrac{1}{3} + \eta\right], \quad J_1 = \left[\tfrac{2}{3} - \eta, \tfrac{2}{3} + \eta\right].$$

Let $U_0 := I_0 \cup J_0$ and $U_1 := I_1 \cup J_1$. Define anchors $a_0 = \tfrac{1}{6}$, $a_1 = \tfrac{5}{6}$. Define $f_0, f_1 \in C([0, 1], [0, 1])$ by

$$f_0(x) = \begin{cases} 3x & x \in I_0 \\ \text{linear interpolation} & x \in J_0 \\ a_0 & x \in [\tfrac{1}{3} + \eta, 1] \end{cases}, \quad f_1(x) = \begin{cases} 3x - 2 & x \in I_1 \\ \text{linear interpolation} & x \in J_1 \\ a_1 & x \in [0, \tfrac{2}{3} - \eta]. \end{cases}$$

Then:

- On its own chamber $U_i$, $f_i$ is (piecewise) expansive with slope $3 > 1$ on $I_i$.

- Off $U_i$, $f_i$ is constant (reset) up to a thin collar.

- Because the images $f_i(U_i)$ cover long subintervals $[0, 1 - 3\eta]$ and $[3\eta, 1]$, each meets both $U_0$ and $U_1$ when $\eta < 1/6$.

- The chambers are separated by $\mathrm{dist}(U_0, U_1) = \Delta = \tfrac{1}{3} - 2\eta > 0$, and $|a_0 - a_1| = 2/3$.

The ping–pong assumptions (PP1)–(PP4) hold, so Condition E2 yields: with $\delta := \min\{\Delta, |a_0 - a_1|\} = \tfrac{1}{3} - 2\eta$ and any $\varepsilon < \delta/2$,

$$N\big(B(k, \{f_0, f_1\}), d_\infty, \varepsilon\big) \geq 2^k.$$

*Example* 11 (Ping–Pong over Subshift (Compact, Expansive)). Fix an alphabet $\mathcal{A} = [r] \cup \{\bullet\}$ with $r \geq 2$ *active* symbols $[r]$ and one extra *padding* symbol $\bullet$. Let

$$\mathcal{X} = \mathcal{A}^{\mathbb{N}} = \{x = (x_0, x_1, x_2, \dots) \mid x_j \in \mathcal{A}\}$$

be the (one-sided) full shift. Equip $\mathcal{X}$ with the standard ultrametric $d_\theta$ for some fixed $\theta \in (0, 1)$:

$$d_\theta(x, y) = \begin{cases} 0 & x = y \\ \theta^{\min\{n \geq 0 | x_n \neq y_n\}} & x \neq y \end{cases}$$

Then $(\mathcal{X}, d_\theta)$ is compact, totally disconnected, and the left shift $\sigma(x)_n = x_{n+1}$ is $L$-Lipschitz with $L = \theta^{-1} > 1$ (hence expansive).

For each $a \in \mathcal{A}$, write the clopen 1-cylinder $[\![a]\!] = \{x \in \mathcal{X} \mid x_0 = a\}$, and the *anchor* sequence $\bar{a} = (a, a, a, \dots)$.

Define $F = \{f_1, \dots, f_r\} \subset C(\mathcal{X}, \mathcal{X})$ by

$$f_a(x) = \begin{cases} \sigma(x) & x \in [\![a]\!] \\ \bar{a} & x \notin [\![a]\!] \end{cases} \qquad a \in [r].$$

Because $[\![a]\!]$ is clopen, each $f_a$ is continuous; on $[\![a]\!]$ it is $\sigma$ (Lipschitz constant $\theta^{-1} > 1$), and on $\mathcal{X} \setminus [\![a]\!]$ it is constant. Thus each $f_a$ is *expansive* (uniform Lipschitz with constant $\theta^{-1} > 1$).

Let $U_a := [\![a]\!]$ (clopen chambers), $a \in [r]$. Then:

- Pairwise separation. If $a \neq b$, then for any $x \in U_a$, $y \in U_b$, $\min d_\theta(x, y) = \theta^0 = 1$. Hence $\mathrm{dist}(U_a, U_b) = 1$.

- Expansion/coverage on own domain. For each $a$, $f_a|_{U_a} = \sigma$ maps $U_a$ bijectively onto $\mathcal{X}$ (surjective and expanding).

- Reset off domain. On $\mathcal{X} \setminus U_a$, $f_a = \mathrm{const}_{\bar{a}}$ (exact reset; the "reset diameter" is 0).

- Separated anchors. $d_\theta(\bar{a}, \bar{b}) = 1$ for $a \neq b$.

These are exactly the ping–pong assumptions (PP1)–(PP4) in Condition E2 with

$$\Delta = \min_{a \neq b} \mathrm{dist}(U_a, U_b) = 1, \qquad \delta_0 = 0, \qquad \min_{a \neq b} d_\theta(\bar{a}, \bar{b}) = 1.$$

Hence our separation scale can be taken as $\delta = 1$.

Let a word $u = a_k \cdots a_1 \in [r]^k$. Define the *tail-padded probe*

$$x_u := \big(a_1, a_2, \ldots, a_k, \underbrace{\bullet, \bullet, \bullet, \ldots}_{\text{all } \bullet}\big) \in \mathcal{X}.$$

Then, by construction,

$$f_u(x_u) = \sigma^k(x_u) = \bar{\bullet},$$

because each of the first $k$ steps sees the correct chamber and applies $\sigma$, peeling off the $k$-letter prefix and revealing the all-$\bullet$ tail.

If $v = b_k \cdots b_1 \neq u$, let $t$ be the *rightmost* index with $a_t \neq b_t$. When applying $f_v$ to $x_u$, the first $t-1$ letters match and act as shifts; at step $t$ we apply $f_{b_t}$ to a sequence whose 0-coordinate is $a_t \neq b_t$, so $f_{b_t}$ resets to the anchor $\bar{b}_t$. From then on, the state never contains $\bullet$ at the 0-coordinate (subsequent resets only use anchors $\bar{b}_s$ with $b_s \in [r]$, and $\sigma$ preserves 0-coordinate $b_s$ on the constant sequence $\bar{b}_s$). Consequently

$$f_v(x_u) \neq \bar{\bullet}.$$

Therefore

$$d_\theta\big(f_u(x_u), f_v(x_u)\big) = d_\theta\big(\bar{\bullet}, f_v(x_u)\big) = 1.$$

This shows *freeness*: distinct words $u \neq v$ define distinct maps $f_u \neq f_v$.

Fix $k \in \mathbb{N}$ and consider the finite probe set

$$P_k := \{x_u \mid u \in [r]^k\} \subset \mathcal{X},$$

of cardinality $r^k$. For any distinct $u, v$ we have just seen that at the coordinate $x_u$,

$$d_\theta\big(f_u(x_u), f_v(x_u)\big) = 1 \geq \delta.$$

Thus the $r^k$ vectors $E_{P_k}(f_u)_{|u|=k} \subset \mathcal{X}^{P_k}$ are pairwise 1-separated in the max metric. By Lemma 7, for every $\varepsilon < \frac{1}{2}$,

$$N\big(B(k, F), d_\infty, \varepsilon\big) \geq r^k.$$

### E.4.3  E3. UNIFORM EXPANSION $\implies$ SUPER-/DOUBLE-EXPONENTIAL LOWERBOUNDS IN $k$

**Condition E3** (Uniformly Expansive Semigroup Grows Super-/Double Exponentially). *Assume:*

    *1. (reset map) there exist a compact $K \subset \mathcal{X}$ and $r \in B(C_r, F)$ and $r(\mathcal{X}) \subset K$.*

    *2. (uniform expanding maps) there exists $A \in B(C_A, F)$ and*

$$\mathrm{coLip}_K(A) \geq \lambda > 1, \qquad \mathrm{Lip}_K(A) \leq \Lambda < \infty.$$

    *3. (writers) there exists a compact set $G \subset B(C_G, F) \cap C(K, K)$ with $\sup_{g \in G} \mathrm{Lip}_K(g) \leq 1$.*

*Fix $\varepsilon > 0$ and $k \in \mathbb{N}$. Consider words of the form*

$$W_k = \{w = A \circ g_k \circ A \circ g_{k-1} \circ \cdots \circ A \circ g_1 \circ r \mid g_j \in G\}.$$

*Then $W_k$ is a subset of $B(Ck + C_r, F)$ with $C = C_A + C_G$, and*

$$N\left(W_k, d_\infty, \varepsilon\right) \geq N\left(B(Ck+O(1), F), d_\infty, \varepsilon\right) \geq \prod_{j=1}^{k} N\left(G, \frac{\varepsilon}{\lambda^{k-j+1}}\right).$$

*Proof.* Let $\delta_j := \varepsilon/\lambda^{k-j+1}$. Choose for each $j$ a $\delta_j$-net $\mathcal{N}_j \subset G$ of maximal cardinality $N(G, \delta_j)$. Consider the set of words $w$ obtained by picking $g_j \in \mathcal{N}_j$ independently. If two such words $w, w'$ first differ at position $j$ (counting from the right), then for all $x \in \mathcal{X}$,

$$d\left((g_j \circ r)(x), (g_j' \circ r)(x)\right) \geq \delta_j$$

after evaluating inside $K$, and subsequent composition with $A$ at least multiplies distances by $\lambda^{k-j+1}$. The 1-Lipschitz $g_\ell$ does not increase distances, so overall

$$d_\infty(w, w') \geq \lambda^{k-j+1} \delta_j = \varepsilon.$$

Thus we obtain a packing of size $\prod_j N(G, \delta_j)$ at scale $\varepsilon$, and the claim follows from $N \geq M$. $\square$

Condition E3 converts information about the layer-wise covering numbers $N(G, \cdot)$ into a covering lower bound for deep compositions. Two regimes are of special interest.

**Condition E3a** (Super-exponential Growth for Finite-dimensional Families). *Suppose $G$ is contained in a $D$-dimensional $C^0$ submanifold of $C(K, K)$ and $\sup_{g \in G} \operatorname{Lip}_K(g) \leq 1$. Then there is a constant $c_1 > 0$ such that for all sufficiently small $\delta > 0$,*

$$N(G, d_\infty, \delta) \geq c_1 \delta^{-D}.$$

*Consequently, for every fixed $\varepsilon > 0$,*

$$\log N\left(B(Ck+O(1), F), d_\infty, \varepsilon\right)$$
$$\geq \sum_{j=1}^k \left(D(k-j+1)\log\lambda + D\log(1/\varepsilon) - c_2\right)$$
$$= \frac{D}{2}(\log\lambda)k(k+1) + Dk\log(1/\varepsilon) + O(k).$$

*In particular,*

$$N\left(B(Ck+O(1), F), d_\infty, \varepsilon\right) \geq \lambda^{cDk^2}\varepsilon^{-Dk} = \exp(\Theta(k^2 + k\log(1/\varepsilon)))$$

*for some $c > 0$: growth is super-exponential in $k$.*

*Proof.* The entropy estimate $N(G, \delta) \geq c_1 \delta^{-D}$ is standard for compact $D$-dimensional submanifolds of a Banach space under the $C^0$-norm. Apply Condition E3 and sum the geometric-arithmetic progression. $\square$

**Condition E3b** (Double-exponential Growth for Hölder Balls). *Suppose $\mathcal{X}$ is a $d$-dimensional manifold with bi-Lipschitz coordinates, and $K \subset \mathcal{X}$ be compact. Let*

$$G = \{g \in C(K, K) \mid g = \operatorname{id}_K + u, \|u\|_\infty \leq 1, [u]_{C^\alpha} \leq 1\}$$

*with $\alpha \in (0, 1]$. Then there exists $c > 0$ such that for $\delta \in (0, \delta_0]$,*

$$\log N(G, d_\infty, \delta) \geq c\delta^{-d/\alpha}.$$

*Hence for every fixed $\varepsilon > 0$,*

$$\log N\left(B(Ck+O(1), F), d_\infty, \varepsilon\right) \geq \sum_{j=1}^k c\left(\frac{\lambda^{k-j+1}}{\varepsilon}\right)^{d/\alpha} \geq c'\varepsilon^{-\frac{d}{\alpha}}\lambda^{\frac{d}{\alpha}k},$$

*and therefore*

$$N\left(B(Ck+O(1), F), d_\infty, \varepsilon\right) \geq \exp\left(C\lambda^{\frac{d}{\alpha}k}\varepsilon^{-\lambda^{\frac{d}{\alpha}}}\right) = \exp\left(\exp\left(\Theta(k + \log(1/\varepsilon))\right)\right).$$

*This is double-exponential growth in $k$. (the exponent itself grows exponentially in $k$).*

*Proof.* The entropy bound for Hölder balls in the sup norm is classical and follows by dyadic partitioning of $K$ at mesh size $\delta^{1/\alpha}$, prescribing values on the grid with resolution $\delta$, and using the Hölder constraint to extend; Then apply Condition E3 and sum a geometric series. $\square$

*Example* 12 (Hölder Balls on Euclidean Space). Let $\mathcal{X} = \mathbb{R}^d$ with the bounded metric $d_b$. Fix $K = \overline{B(0,1)}$.

- (Reset) $r(x) := x/\max\{1, |x|\}$ maps $\mathbb{R}^d$ into $K$, continuously.

- (Expansion) pick $\lambda > 1$ and define in polar coordinates an $A$ that equals $x \mapsto \lambda x$ on $\overline{B(0, 1/2)}$, smoothly interpolates on the annulus to the identity, and equals the identity outside $B(0,1)$. Then $\mathrm{coLip}_K(A) \geq \lambda' > 1$ for some $\lambda'$ close to $\lambda$.

- (Writers) take $G$ to be the $C^\alpha$-Hölder ball of small perturbations supported in $B(0, 1/2)$, with $\|u\|_\infty \leq 1$ and $[u]_{C^\alpha} \leq 1$, and define $g = \mathrm{id} + u$. Each $g$ is 1-Lipschitz on $K$ after shrinking the ball if needed.

All three ingredients lie in a compact $F := \overline{\{r, A\} \cup G}$. Condition E3b applies and yields a double-exponential lower bound.

### E.4.4 EXPONENTIAL UPPERBOUNDS

Let $L := \sup_{f \in F} \mathrm{Lip}(f)$. A standard chaining argument gives

$$N\left(B(k, F), d_\infty, \varepsilon\right) \leq \prod_{j=0}^{k-1} N\left(F, \frac{\varepsilon}{L^j}\right).$$

Indeed, approximate the rightmost factor in a word within $\varepsilon/L^{k-1}$, the next within $\varepsilon/L^{k-2}$, and so on; the Lipschitz constants propagate the errors to at most $\varepsilon$ at the output. This complements Condition E3, which provides a matching-flavor lower bound under expansion.

## F PROOFS FOR BALANCING BIAS-VARIANCE TRADE-OFF IN DEPTH

Throughout, we minimize the upper bound

$$\mathsf{gen}(k, n) \lesssim \mathsf{bias}(k) + \mathsf{var}(k, n),$$

treat $k$ as a positive real (round to the nearest integer at the end), and use the standard heuristic that—because $\mathsf{bias}(k)$ is decreasing in $k$ while $\mathsf{var}(k, n)$ is increasing—the minimizer occurs where the two terms are of the same order:

$$\mathsf{bias}(k^*) \asymp \mathsf{var}(k^*, n).$$

Solving that equation gives $k^*$; plugging back yields the minimized rate. (If a term does not cross, the optimum is at a boundary, but in all four regimes below they do cross for large $n$.)

### F.1 EP (EXP-DECAY BIAS, POLY-GROWTH VARIANCE)

$$\mathsf{bias}(k) = e^{-\alpha k}, \qquad \mathsf{var}(k, n) = n^{-1/2} k^{\gamma/2}.$$

Balance:
$$e^{-\alpha k} \asymp n^{-1/2} k^{\gamma/2} \quad \Longleftrightarrow \quad \alpha k = \tfrac{1}{2} \log n - \tfrac{\gamma}{2} \log k.$$

As $n \to \infty$, $\log k \ll \log n$, so an asymptotic solution is

$$k^* = \frac{1}{2\alpha} \Big( \log n - \gamma \log \log n + O(1) \Big).$$

Plugging back (either term) gives

$$\mathsf{gen}(k^*, n) \asymp n^{-1/2} (\log n)^{\gamma/2} \quad \text{(more precisely } \approx (2\alpha)^{-\gamma/2} n^{-1/2} (\log n)^{\gamma/2} \text{ up to a factor} \asymp 1).$$

### F.2 EL (EXP-DECAY BIAS, LOG-GROWTH VARIANCE)

$$\mathsf{bias}(k) = e^{-\alpha k}, \qquad \mathsf{var}(k, n) = \sqrt{\log k / n}.$$

Balance:

$$e^{-\alpha k} \asymp \sqrt{\log k / n} \quad \Longleftrightarrow \quad \alpha k = \tfrac{1}{2} \log n - \tfrac{1}{2} \log \log k.$$

Hence

$$k^* = \frac{1}{2\alpha} \Big( \log n - \log \log \log n + o(1) \Big),$$

and

$$\mathsf{gen}(k^*, n) \asymp \sqrt{\log \log n / n}.$$

### F.3 PP (POLY-DECAY BIAS, POLY-GROWTH VARIANCE)

$$\mathsf{bias}(k) = k^{-\beta}, \qquad \mathsf{var}(k, n) = n^{-1/2} k^{\gamma/2}.$$

Balance:

$$k^{-\beta} \asymp n^{-1/2} k^{\gamma/2} \quad \Longleftrightarrow \quad k^{\beta + \gamma/2} \asymp n^{1/2}.$$

Thus

$$k^* \asymp n^{1/(2\beta + \gamma)}, \qquad \mathsf{gen}(k^*, n) \asymp n^{-\beta/(2\beta + \gamma)}.$$

### F.4 PL (POLY-DECAY BIAS, LOG-GROWTH VARIANCE)

$$\mathsf{bias}(k) = k^{-\beta}, \qquad \mathsf{var}(k, n) = \sqrt{\log k / n}.$$

Balance (square both sides):

$$k^{-2\beta} \asymp \frac{\log k}{n} \quad \Longleftrightarrow \quad k^{2\beta} \log k \asymp n.$$

Let $k = e^t$. Then $t e^{2\beta t} \asymp n$, so

$$2\beta t = W(2\beta n) \quad \Longrightarrow \quad k^* = \exp\Big(\frac{1}{2\beta} W(2\beta n)\Big) = \Big(\frac{2\beta n}{W(2\beta n)}\Big)^{1/(2\beta)},$$

where $W$ is the Lambert $W$ function. Consequently,

$$\mathsf{gen}(k^*, n) \asymp \sqrt{\frac{W(2\beta n)}{2\beta n}} \sim \sqrt{\frac{\log n}{2\beta n}} \quad (\text{since } W(x) \sim \log x).$$

## G NOTES ON RADEMACHER COMPLEXITY AND GENERALIZATION ERROR BOUNDS

We refer to (Martin J. Wainwright, 2019; Mohri et al., 2018) for more details on Rademacher complexity and its application to generalization error bounds.

### G.1 RADEMACHER COMPLEXITY

**Sample space** Let $(\mathcal{Z}, \mathcal{B}(\mathcal{Z}))$ be a measurable space that serves as the sample space, and let $P$ be an unknown Borel probability measure on $\mathcal{B}(\mathcal{Z})$. We observe an i.i.d. sample $S = (Z_1, \ldots, Z_n) \overset{\text{iid}}{\sim} P$.

**Hypothesis class** A hypothesis is a measurable function $h : \mathcal{Z} \to \mathbb{R}$. The hypothesis class is a separable set $\mathcal{H} \subset \{h : \mathcal{Z} \to \mathbb{R}\}$. Here, separability is assumed to prevent pathologies where the supremum over uncountable $\mathcal{H}$ is non-measurable.

**Rademacher complexities** Introduce an independent sequence of Rademacher variables $\sigma = (\sigma_1, \ldots, \sigma_n)$ with $\Pr(\sigma_i = 1) = \Pr(\sigma_i = -1) = \frac{1}{2}$. For the fixed sample $S$ the empirical Rademacher complexity of $\mathcal{H}$ is

$$\hat{\mathfrak{R}}_S(\mathcal{H}) = \mathbb{E}_\sigma\left[\sup_{h \in \mathcal{H}} \frac{1}{n} \sum_{i=1}^n \sigma_i h(Z_i)\right].$$

Averaging this quantity over all samples of size $n$ drawn from $P$ gives the (distribution-dependent) Rademacher complexity

$$\mathfrak{R}_n(\mathcal{H}) = \mathbb{E}_{S \overset{\text{iid}}{\sim} P}[\hat{\mathfrak{R}}_S(\mathcal{H})].$$

**Properties**

- **Symmetrization**: $\mathbb{E}_S\left[\sup_{h \in \mathcal{H}}(P - \hat{P})h\right] \leq 2\mathfrak{R}_n(\mathcal{H})$ (Bartlett & Mendelson, 2002)
- **Ledoux–Talagrand contraction lemma**: if $\psi : \mathbb{R} \to \mathbb{R}$ is $\beta$-Lipschitz and $\psi(0) = 0$ then $\hat{\mathfrak{R}}_S(\psi \circ \mathcal{H}) \leq \beta\hat{\mathfrak{R}}_S(\mathcal{H})$ (Mohri et al., 2018, Lemma 5.7)
- **Monotonicity** and **convexity preservation**: if $\mathcal{H} \subset \mathcal{H}'$ or $\text{conv}(\mathcal{H})$ is taken, complexity cannot increase (Bartlett & Mendelson, 2002)

G.2 UNIFORM DEVIATION FOR LIPSCHITZ, BOUNDED LOSS

**Sample space** Let $\mathcal{X}$ be a measurable space. Take an input-output sample space $\mathcal{X} \times \mathcal{Y}$ with $\mathcal{Y} = \mathbb{R}$.

**Hypothesis class** Let $\mathcal{H} \subset \{h : \mathcal{X} \to \mathbb{R}\}$ be any class of measurable functions.

**Loss function** Fix a measurable loss $\ell : \mathbb{R} \to [0, b]$ that is $\beta_\ell$-Lipschitz in its *first* argument:
$$|\ell(a, y) - \ell(a', y)| \leq \beta_\ell |a - a'| \qquad (a, a' \in \mathbb{R}, y \in \mathbb{R}).$$
For brevity write $\ell \cdot h(x, y) = \ell(h(x), y)$ and $\ell \cdot \mathcal{H} = \{\ell \cdot h : h \in \mathcal{H}\} \subset [0, b]^{\mathcal{X} \times \mathcal{Y}}$.

**Risks** Given a sample $S = ((X_i, Y_i))_{i=1}^n$, define empirical and population risks

$$\hat{L}_n[h] = \frac{1}{n} \sum_{i=1}^n \ell(h(X_i), Y_i), \qquad L[h] = \mathbb{E}_{(X,Y) \sim P}[\ell(h(X), Y)].$$

**Theorem 5** (uniform deviation theorem (in expectation)).

$$\mathbb{E}_S\left[\sup_{h \in \mathcal{H}}|L[h] - \hat{L}_n[h]|\right] \leq 2\beta_\ell\mathfrak{R}_n(\mathcal{H}). \tag{33}$$

*Proof.* The symmetrization identity applied to $\ell \cdot \mathcal{H}$ gives

$$\mathbb{E}_S\left[\sup_{h \in \mathcal{H}}(L[h] - \hat{L}_n[h])\right] \leq 2\mathfrak{R}_n(\ell \cdot \mathcal{H}). \tag{34}$$

Using the contraction lemma with scale $\beta_\ell$ and the fact $\ell(a, y) - \ell(0, y) = \psi_y(a)$ has Lipschitz constant $\beta_\ell$, one gets

$$\mathfrak{R}_n(\ell \cdot \mathcal{H}) \leq \beta_\ell\mathfrak{R}_n(\mathcal{H}). \tag{35}$$

Combining Eqs. (34) and (35) yields the assertion. $\square$

**Theorem 6** (uniform deviation theorem (high probability)). *With probability at least $1 - \delta$ over the i.i.d. draw $S \sim P^n$,*

$$\sup_{h \in \mathcal{H}}|L[h] - \hat{L}_n[h]| \leq 2\beta_\ell\hat{\mathfrak{R}}_S(\mathcal{H}) + b\sqrt{\frac{2\log(1/\delta)}{n}} \tag{36}$$

$$\sup_{h \in \mathcal{H}}|L[h] - \hat{L}_n[h]| \leq 2\beta_\ell\mathfrak{R}_n(\mathcal{H}) + b\sqrt{\frac{2\log(1/\delta)}{n}} \tag{37}$$

*Proof.* Define $G(S) = \sup_{h \in \mathcal{H}} |L[h] - \hat{L}_n[h]|$. Replacing one coordinate $(X_j, Y_j)$ in $S$ can change each summand by at most $b/n$, so

$$|G(S) - G(S')| \leq \frac{b}{n} \qquad \text{whenever } S, S' \text{ differ in exactly one entry.}$$

Hence $G$ satisfies the bounded-difference condition and McDiarmid's inequality gives

$$\Pr\left\{ G(S) \geq \mathbb{E}[G(S)] + \sqrt{\frac{2b^2 \log(1/\delta)}{n}} \right\} \leq \delta.$$

Inserting Eq. (33) and simplifying constants we obtain, with probability $1 - \delta$,

$$\sup_{h \in \mathcal{H}} |L[h] - \hat{L}_n[h]| \leq 2\beta_\ell \hat{\mathfrak{R}}_S(\mathcal{H}) + b\sqrt{\frac{2\log(1/\delta)}{n}}. \tag{38}$$

Taking expectation and applying Jensen's inequality yields

$$\sup_{h \in \mathcal{H}} |L[h] - \hat{L}_n[h]| \leq 2\beta_\ell \mathfrak{R}_n(\mathcal{H}) + b\sqrt{\frac{2\log(1/\delta)}{n}}, \tag{39}$$

with the same probability. $\qquad\square$

The bound controls the worst-case *generalization gap* uniformly over $\mathcal{H}$. If $\hat{\mathfrak{R}}_S(\mathcal{H})$ (data-dependent) or $\mathfrak{R}_n(\mathcal{H})$ (distribution-dependent) is small—scaling, say, like $O(1/\sqrt{n})$—then empirical risk minimization on $S$ provably finds a hypothesis whose population risk is close to optimal.

$\hat{\mathfrak{R}}_S$ quantifies how well the function class can *correlate with random noise* on the sample. A low value implies strong regularization, yielding smaller sample complexity. The second term is a universal concentration penalty that decays at the Monte-Carlo rate $O(\sqrt{\log(1/\delta)/n})$.

### G.3   BOUNDING RADEMACHER COMPLEXITY

Let $S = (X_1, \ldots, X_n)$ be an i.i.d. sample from $P$. Throughout this section every $h \in \mathcal{H}$ is assumed to satisfy $\|h\|_\infty \leq b$ for some constant $b > 0$.

**Massart's finite class lemma**

**Theorem 7** (Massart's finite class lemma). *Suppose $\mathcal{H}$ is a finite family, $|\mathcal{H}| < \infty$. Then,*

$$\hat{\mathfrak{R}}_S(\mathcal{H}) \leq b\sqrt{\frac{2\log|\mathcal{H}|}{n}}. \tag{40}$$

We refer to (Mohri et al., 2018, Theorem 3.7) for the proof.

The bound depends logarithmically on the cardinality of the class, illustrating how modeling with a finite but exponentially large dictionary can still be statistically benign.

**Covering numbers and Dudley's entropy integral**   When $\mathcal{H}$ is infinite a combinatorial bound like Eq. (40) is no longer adequate. A refined control is obtained by chaining the increments of the empirical process in the empirical $L^2$-metric $\|h - h'\|_S = \left( \frac{1}{n} \sum_{i=1}^n \left( h(X_i) - h'(X_i) \right)^2 \right)^{1/2}$.

For any $\varepsilon > 0$ let $N(\varepsilon, \mathcal{H}, \|\cdot\|_S)$ denote the minimal number of $\|\cdot\|_S$-balls of radius $\varepsilon$ needed to cover $\mathcal{H}$.

**Theorem 8** (Dudley's entropy integral inequality).

$$\hat{\mathfrak{R}}_S(\mathcal{H}) \leq \frac{12}{\sqrt{n}} \int_0^{\text{diam}(\mathcal{H}, \|\cdot\|_S)} \sqrt{\log N(\varepsilon, \mathcal{H}, \|\cdot\|_S)}\, d\varepsilon. \tag{41}$$

We refer to (Martin J. Wainwright, 2019, Theorem 5.22) for the proof.

**Consequences and examples** *For Hölder-smooth functions.* If $\mathcal{H}$ is a unit ball of a Hölder class of order $s > d/2$ on $[0,1]^d$, classical approximation theory gives $\log N(\varepsilon, \mathcal{H}, \|\cdot\|_S) \lesssim \varepsilon^{-d/s}$. Inserting this into Eq. (41) yields

$$\hat{\mathfrak{R}}_S(\mathcal{H}) \lesssim \frac{1}{\sqrt{n}} \int_0^1 \varepsilon^{-d/(2s)} d\varepsilon = \frac{C_{d,s}}{n^{s/(2s+d)}}.$$

Thus Rademacher complexity reproduces the minimax rate $n^{-2s/(2s+d)}$ for non-parametric regression.

*For linear prediction in $\mathbb{R}^p$.* With $\mathcal{H} = \{x \mapsto \langle w, x \rangle : \|w\|_2 \leq 1\}$ and $\|X_i\|_2 \leq 1$, the covering number in $\|\cdot\|_S$ is bounded by $N(\varepsilon) \leq (3/\varepsilon)^p$. Dudley's integral then recovers $\hat{\mathfrak{R}}_S(\mathcal{H}) \leq \sqrt{p/n}$, i.e. the familiar parametric rate.

Dudley's entropy integral Eq. (41) and Massart's finite-class lemma Eq. (40) together form a bridge between combinatorial and geometric measures of capacity: the first quantifies richness by counting distinguishable hypotheses at each scale, the second by raw cardinality. In practice one often proves (data-dependent) covering-number bounds and plugs them into Eq. (41); the resulting Rademacher complexity then feeds directly into high-probability risk bounds through the symmetrization and concentration arguments.

# H  NOTES ON COVERING AND PACKING NUMBERS

## H.1  PSEUDO-METRIC SPACE

**Definition 1** (pseudo-metric space)**.** A pseudo-metric space $(X, \rho)$ is a pair of set $X$ and non-negative real-valued function $\rho : X \times X \to \mathbb{R}_{\geq 0}$ (called pseudo-metric) satisfying the following three axioms:

- $\forall x, y \in X : \quad \rho(x, y) = \rho(y, x)$

- $\forall x, y, z \in X : \quad \rho(x, z) \leq \rho(x, y) + \rho(y, z)$

- $\forall x \in X : \rho(x, x) = 0$.

Note that it is a metric space if it further satisfies

- $\forall x, y \in X : \rho(x, y) = 0 \implies x = y$.

In other words, in a pseudo-metric space, two distinct points $x, y \in X$ may achieve zero distance $\rho(x, y) = 0$.

## H.2  COVERING AND PACKING NUMBERS

Let $(X, \rho)$ be a pseudo-metric space. Let $B(x; \varepsilon) := \{y \in X \mid \rho(x, y) \leq \varepsilon\}$ denote the closed ball in $X$ of radius $\varepsilon$ centered at a point $x \in X$.

**Definition 2.** A subset $C \subset X$ is a $\varepsilon$-covering if for every $x \in X$ there exists $y \in C$ satisfying $\rho(x, y) \leq \varepsilon$. In other words,

$$X \subset \bigcup_{x \in C} B(x; \varepsilon).$$

The minimal cardinality of $\varepsilon$-covering, i.e.

$$N(X, \rho, \varepsilon) := \inf \{|C| \mid C \subset X \text{ is } \varepsilon\text{-covering}\}$$

is called the covering number of $X$.

**Definition 3.** A subset $P \subset X$ is an $\varepsilon$-packing if for every $x, y \in P$ $\rho(x, y) > \varepsilon$. In other words,

$$\forall x, y \in P, x \neq y \implies B(x; \varepsilon) \cap B(y; \varepsilon) = \emptyset$$

The maximal cardinality of $\varepsilon$-packing, i.e.

$$M(X, \rho, \varepsilon) := \sup \{|P| \mid P \subset X \text{ is } \varepsilon\text{-packing}\}$$

is called the packing number of $X$.

While the covering number involves a minimization problem, the packing number involves a maximization problem. The following theorem relates the duality of two quantities.

**Theorem 9.** *Let $(X, \rho)$ be a pseudo-metric space. For any positive number $\varepsilon > 0$, we have*

$$N(X, \rho, \varepsilon) \leq M(X, \rho, \varepsilon) \leq N(X, \rho, \varepsilon/2).$$

### H.3  COVERING NUMBER OF COMPOSITIONS

Let $(X, d_X), (Y, d_Y), (Z, d_Z)$ be pseudo-metric spaces. Let $C(X, Y), C(Y, Z), C(X, Z)$ denote the complete metric spaces of continuous maps equipped with uniform metrics:

$$d_{C(X,Y)}(f, f') := \sup_{x \in X} d_Y(f(x), f'(x)), \qquad\qquad f, f' \in C(X, Y)$$

$$d_{C(Y,Z)}(g, g') := \sup_{y \in Y} d_Z(g(y), g'(y)), \qquad\qquad g, g' \in C(Y, Z)$$

$$d_{C(X,Z)}(h, h') := \sup_{x \in X} d_Z(h(x), h'(x)), \qquad\qquad h, h' \in C(X, Z)$$

respectively. Given a subclasses of continuous maps

$$F \subset C(X, Y), \qquad G \subset C(Y, Z),$$

define their composition class

$$G \circ F := \{g \circ f \mid g \in G, f \in F\} \subset C(X, Z).$$

**Lemma 8** (composition lemma)**.** *Assume every $g \in G$ is at most $L$-Lipschitz: There exists $L > 0$ for every $g \in G$ such that*

$$d_Z\big(g(y), g(y')\big) \leq L\, d_Y(y, y') \qquad (\forall y, y' \in Y).$$

*Then for any positive numbers $\varepsilon, \delta_G, \delta_F > 0$ satisfying $\varepsilon = \delta_G + L\delta_F$,*

$$N\Big(\varepsilon, G \circ F, d_{C(X,Z)}\Big) \leq N\Big(\delta_G, G, d_{C(Y,Z)}\Big) N\Big(\delta_F, F, d_{C(X,Y)}\Big) \tag{42}$$

*Taking logs gives a chain rule of metric entropies:*

$$\log N(\varepsilon, G \circ F) \leq \log N(\delta_G, G) + \log N\big(\delta_F, F\big). \tag{43}$$

*Proof.* Build nets for $G$ and $F$. Choose

$$G_{\text{net}} := \{g_1, \ldots, g_M\}, \qquad F_{\text{net}} := \{f_1, \ldots, f_N\}$$

such that

$$d_{C(Y,Z)}(g, g_m) \leq \delta_G, \quad d_{C(X,Y)}(f, f_n) \leq \delta_F.$$

Bound the composition error. For any $g \in G, f \in F$ pick the nearest net points $g_m, f_n$ and use triangle inequalities:

$$
\begin{aligned}
d_{C(X,Z)}(g \circ f, g_m \circ f_n) &\leq d_{C(X,Z)}(g \circ f, g \circ f_n) + d_{C(X,Z)}(g \circ f_n, g_m \circ f_n) \\
&\leq L d_{C(X,Y)}(f, f_n) + d_{C(Y,Z)}(g, g_m) \\
&\leq L\delta_F + \delta_G = \varepsilon.
\end{aligned}
$$

Thus the composite error is at most $\varepsilon$.

Counting. The set $G_{\text{net}} \circ F_{\text{net}}$ is an $\varepsilon$-net of size $MN$, yielding the stated bound. $\qquad\square$

## I  NOTES ON TOTALLY BOUNDEDNESS OF FUNCTION SETS AND ARZELÀ–ASCOLI PRINCIPLE

Here we list sufficient conditions for totally boundedness of a set $H \subset C(X)$ in the uniform norm $\|f\|_\infty = \sup_{x \in X} |f(x)|$. Throughout, $X$ is a topological space and

$$C(X) := \{f : X \to \mathbb{R} \mid f \text{ is continuous and bounded}\}, \qquad \|f\|_\infty = \sup_{x \in X} |f(x)|.$$

**Total boundedness.** A subset $H \subset C(X)$ is totally bounded if for every $\varepsilon > 0$ there exist finitely many points $f_1, \ldots, f_N \in C(X)$ such that $H \subset \bigcup_{i=1}^N B_\infty(f_i, \varepsilon)$, where $B_\infty(f, \varepsilon) = \{g \mid \|g - f\|_\infty < \varepsilon\}$.

**Equicontinuity.** A family $H \subset C(X)$ is *equicontinuous* at $x \in X$ if for every $\varepsilon > 0$ there is a neighbourhood $U$ of $x$ with $|f(y) - f(x)| < \varepsilon$ for all $y \in U$ and $f \in H$. It is *equicontinuous on $X$* if this holds at every point.

**Uniform boundedness.** $H$ is uniformly bounded if $\sup_{f \in H} \|f\|_\infty < \infty$.

**Compact exhaustion.** For a locally compact Hausdorff space we write $C_0(X) \subset C(X)$ for the subspace of functions *vanishing at infinity*, i.e. $\forall \varepsilon > 0 \exists$ compact $K \subset X : |f(x)| < \varepsilon$ for all $x \notin K$.

## I.1 A GENERIC ARZELÀ–ASCOLI PRINCIPLE

**Theorem 10** (Abstract Arzelà–Ascoli)**.** *Let $X$ be a* Tychonoff *space (completely regular Hausdorff). A set $H \subset C(X)$ is totally bounded in $\|\cdot\|_\infty$ provided that*

1. *$H$ is uniformly bounded, and*

2. *for every compact $K \subset X$ the restriction $H|_K := \{f|_K \mid f \in H\}$ is equicontinuous on $K$.*

The proof combines (i) uniform boundedness to control the range and (ii) the classical Arzelà–Ascoli theorem on each compact $K$; a diagonal argument then yields a finite $\varepsilon$-net on $X$.

## I.2 CASE-BY-CASE CONDITIONS

### I.2.1 $X$ COMPACT HAUSDORFF

Because $X$ itself is compact, condition (2) above is just *global* equicontinuity. Hence

**Corollary 1** (Classical Arzelà–Ascoli)**.** *If $X$ is compact Hausdorff and $H \subset C(X)$ is* uniformly bounded *and* equicontinuous on $X$, *then $H$ is totally bounded (and its closure is compact in $C(X)$).*

*(For real-valued functions "pointwise relative compactness" in the usual statements reduces to uniform boundedness.)*

### I.2.2 $X$ LOCALLY COMPACT HAUSDORFF

Now $X$ need not be compact. Total boundedness in $C(X)$ fails unless one controls behavior at infinity. A convenient framework is $C_0(X)$.

**Theorem 11** (Arzelà–Ascoli for $C_0(X)$)**.** *Let $X$ be locally compact Hausdorff and let $H \subset C_0(X)$. Suppose*

1. *Uniform boundedness: $\sup_{f \in H} \|f\|_\infty < \infty$.*

2. *Local equicontinuity: for every compact $K \subset X$ the family $H|_K$ is equicontinuous.*

3. *Uniform vanishing at infinity: $\forall \varepsilon > 0 \exists$ compact $K \subset X : \sup_{f \in H} \sup_{x \notin K} |f(x)| < \varepsilon$.*

*Then $H$ is totally bounded in $C_0(X)$.*

Condition (3) ensures that all functions become uniformly small outside a common compact set, enabling one to restrict attention to a compact domain where Corollary 1 applies.

### I.2.3 $X$ METRIC

Let $(X, d)$ be a metric space. Because metric spaces are paracompact and first-countable, local equicontinuity simplifies to the existence of a *uniform modulus of continuity*.

**Theorem 12** (Metric Arzelà–Ascoli). *If there exists a* modulus of continuity $\omega : [0, \infty) \to [0, \infty)$ *with* $\omega(r) \to 0(r \to 0)$ *such that*

$$|f(x) - f(y)| \leq \omega\left(d(x, y)\right) \quad \forall f \in H, \forall x, y \in X,$$

*and $H$ is uniformly bounded, then $H$ is totally bounded in $C(X)$.*

A common—and often sufficient—specialization is a *uniform Lipschitz bound*:

$$\sup_{f \in H} \mathrm{Lip}(f) < \infty, \qquad \mathrm{Lip}(f) = \sup_{x \neq y} \frac{|f(x) - f(y)|}{d(x, y)}.$$

### I.2.4 $X$ PSEUDO-METRIC

A pseudo-metric $d$ allows zeros for $x \neq y$. Let $\tilde{X} = X/\sim$ with $x \sim y \iff d(x, y) = 0$; then $d$ induces a genuine metric $\tilde{d}$ on $\tilde{X}$, and composition with the quotient map identifies $C(\tilde{X}) \cong C(X)$. Apply Theorem 12 to $(\tilde{X}, \tilde{d})$. Explicitly:

**Corollary 2.** *If a modulus of continuity $\omega$ as in Theorem 12 exists* with respect to the pseudo-metric *$d$, and $H$ is uniformly bounded, then $H$ is totally bounded in $C(X)$.*

### I.2.5 WEAKER COMPACTNESS HYPOTHESES ON $H$

Frequently $H$ is known to satisfy a property *weaker* than full relative compactness—e.g. *uniform boundedness plus equicontinuity*, but *without* an a priori modulus common to all $f \in H$. The following lemma fills the gap when $X$ is metric (or after reducing to a metric space as in Appendix I.2.4).

**Lemma 9** (Modulus extraction). *Let $(X, d)$ be a totally bounded metric space and let $H \subset C(X)$ be uniformly bounded and equicontinuous. Then there exists a* single *modulus $\omega$ satisfying the assumption of Theorem 12. Consequently $H$ is totally bounded.*

*Sketch.* Fix a dense sequence $(x_n)$ in $X$. For each $k \in \mathbb{N}$ use equicontinuity at $x_k$ and uniform boundedness to find a local modulus; take a maximum over finitely many $x_k$'s to build a global $\omega$. Details follow standard proofs of the classical Ascoli theorem. $\square$

### I.3 POSITIVE EXAMPLES

*Example* 13 (compact domain). Let $X = [0, 1]$ (compact metric space), and

$$H := \{\text{polynomials } p \text{ of degree} \leq m \text{ such that } |p(x)| \leq 1 \text{ for all } x \in [0, 1]\}.$$

Then, $H$ is totally bounded because all the criteria of Theorem 12 are satisfied as follows:

- Uniform boundedness: $\|p\|_\infty \leq 1$.

- Common Lipschitz constant exists on the compact interval (classical Markov-type estimates).

*Example* 14 (vanishing at infinity). Let $X = \mathbb{R}$, and

$$H = \{x \mapsto \sin(x/n)\}_{n \in \mathbb{N}}.$$

Then $H$ is totally bounded because all conditions of Theorem 11 hold as follows:

- Uniform boundedness: $\|f\|_\infty \leq 1$.

- Common Lipschitz bound: $\mathrm{Lip}(f) \leq 1$.

- Uniformly vanishing at infinity: for any $\varepsilon > 0$ choose compact $K = [-R, R]$ with $R$ large; then $|f(x)| < \varepsilon$ for $|x| > R$ uniformly in $n$.

### I.3.1 FAILURE EXAMPLES

*Example* 15 (due to behavior at infinity). Let $X = \mathbb{R}$ (locally compact but not compact), and

$$H = \{x \mapsto \sin(nx)\}_{n \in \mathbb{N}}.$$

Then, $H$ is *not* totally bounded in $C(\mathbb{R})$ because The functions do *not* vanish at infinity uniformly; Condition (3) of Theorem 11 fails.

*Example* 16 (due to lack of equicontinuity). Let $X = [0, 1]$, and

$$H = \left\{x \mapsto \sqrt{x} + 1/n\right\}_{n \in \mathbb{N}}.$$

Then $H$ is *not* totally bounded because uniform boundedness holds, but the functions are *not* equicontinuous at $x = 0$ (the derivative blows up); Criterion (2) of Corollary 1 fails.

## J  GUIVARC'H–BASS FORMULA AND HOMOGENEOUS DIMENSION

Following Breuillard (2014), we quickly overview homogeneous dimension and Guivarc'h–Bass formula for locally compact groups with polynomial growth.

### J.1  HOMOGENEOUS DIMENSION

**Definition 4** (Homogeneous dimension of nilpotent Lie groups). Let $N$ be a simply connected nilpotent Lie group with Lie algebra $\mathfrak{n}$ and central descending series $C_i(\mathfrak{n})$ (so $C_1(\mathfrak{n}) = \mathfrak{n}$, $C_{i+1}(\mathfrak{n}) = [\mathfrak{n}, C_i(\mathfrak{n})]$). The *homogeneous dimension* of $N$ is

$$d(N) = \sum_{i \geq 1} \dim C_i(\mathfrak{n}).$$

Equivalently, for a grading $\mathfrak{n} = \bigoplus_{i \geq 1} \mathfrak{m}_i$ adapted to the lower central series, one has

$$d(N) = \sum_{i \geq 1} i \cdot \dim \mathfrak{m}_i.$$

The second formula is the Guivarc'h–Bass form; see below.

### J.2  THE GUIVARC'H–BASS FORMULA

**Theorem 13** (Corollary 2.9 and Equation 17). *For a (simply connected) nilpotent Lie group $N$ with a compact neighborhood $\mathcal{U}$ of the identity and any $n \geq 1$, volume of the product sets satisfies*

$$C_1 n^d \leq \mathrm{vol}_N(\mathcal{U}^n) \leq C_2 n^d$$

*for some constants $C_1, C_2 > 0$, where*

$$d = \sum_{i \geq 1} i \dim \mathfrak{m}_i \qquad \text{(Guivarc'h–Bass)}$$

*with $\mathfrak{n} = \bigoplus_{i \geq 1} \mathfrak{m}_i$ a grading compatible with the lower central series; equivalently $d = \sum_{i \geq 1} \dim C_i(\mathfrak{n})$.*

*Remark* 7. In the general case (Theorem 1.1), the exponent $d(G)$ for any locally compact $G$ of polynomial growth is exactly the Guivarc'h–Bass homogeneous dimension of the graded nilpotent Lie group that arises as the asymptotic cone (graded nilshadow) of $G$.

### J.3  HOW HOMOGENEOUS DIMENSION CONTROLS VOLUME GROWTH

**Theorem 14** (General $G$, Theorem 1.1). *$G$ locally compact of polynomial growth, $\Omega$ compact symmetric generating set. Then, $\mathrm{vol}_G(\Omega^n) \sim c(\Omega)n^{d(G)}$ with $d(G)$ equal to the Guivarc'h–Bass homogeneous dimension of the graded nilshadow (asymptotic cone).*

## J.4 Sufficient conditions for polynomial growth

**Connected Lie groups of type** $(R)$**.** A connected Lie group $S$ has polynomial growth iff it is of type $(R)$, i.e. $\mathrm{ad}(x)$ has only purely imaginary eigenvalues for all $x \in \mathfrak{s}$. In particular, such $S$ is solvable-by-compact, and every connected nilpotent Lie group is of type $(R)$, hence of polynomial growth. (Guivarc'h–Jenkins characterization)

**Passing to cocompact subgroups / compact quotients.** Polynomial growth is preserved in both directions when passing to a *cocompact subgroup* and when taking quotients by *compact normal* subgroups. (Lemma 7.7)

**Discrete subgroups inside solvable type** $(R)$**.** If $\Gamma$ is a discrete subgroup of a connected solvable Lie group of type $(R)$, then $\Gamma$ is *virtually nilpotent* (hence of polynomial growth). (Remark 7.8)

**Virtually nilpotent (Gromov, 1981; Losert, 1987)** Finitely generated groups of polynomial growth are *virtually nilpotent*. Losert extends the structural picture to locally compact groups.

These give practical sufficient hypotheses: *nilpotent* (or graded nilpotent/Carnot), *connected type* $(R)$, *cocompact embedding in a polynomial-growth group*, and *virtually nilpotent discrete subgroups* of solvable type $(R)$.

## J.5 Concrete examples of homogeneous dimension

The homogeneous dimension $d(G)$ coincides with the algebraic/geometric dimension $d$ if the group is abelian (e.g. $\mathbb{Z}^d$ and $\mathbb{R}^d$); while $d(G)$ is larger if the group is non-abelian. Thus the volume growth faster when $G$ is non-abelian.

*Example* 17 ($\mathbb{Z}^d$ and $\mathbb{R}^d$ (abelian)). Grading has only layer $\mathfrak{m}_1$ of dimension $d$; thus $d(N) = 1 \cdot d = d$. Hence volume grows like $t^d$. general formula above.)

*Example* 18 (Heisenberg group $H_3$, Section 9). $\dim \mathfrak{m}_1 = 2$ and $\dim \mathfrak{m}_2 = 1$, so $d(H_3) = 1 \cdot 2 + 2 \cdot 1 = 4$.

*Example* 19 (Heisenberg group $H_5$, Section 9.2). $\dim \mathfrak{m}_1 = 4$, $\dim \mathfrak{m}_2 = 1$, so $d(H_5) = 4 + 2 = 6$.

*Example* 20 (Unitriangular group $UT(n)$ — strictly upper triangular $n \times n$ matrices, step $n - 1$). Using Guivarc'h–Bass, the $i$-th layer has $\dim \mathfrak{m}_i = n - i$ (entries on the $i$-th superdiagonal). Hence

$$d(UT(n)) = \sum_{i=1}^{n-1} i(n - i) = \frac{n(n-1)(n+1)}{6}.$$

*Example* 21 (A worked solvable example, Example 3.3). For $G = \mathbb{R} \ltimes_\varphi \mathbb{R}^n$ where the unipotent part of $\varphi_t$ has $n_k$ Jordan blocks of size $k$, the paper computes

$$d(G) = 1 + \sum_{k \geq 1} \frac{k(k+1)}{2} n_k.$$