# OpenReview forum: "Why and When Deep is Better than Shallow: An Implementation-Agnostic State-Transition View of Depth Supremacy"
_ICLR.cc/2026/Conference — Submitted to ICLR 2026_

### Official Review · Reviewer_Z8EL · 2025-10-19

**Soundness:** 3
**Presentation:** 3
**Contribution:** 2
**Rating:** 4
**Confidence:** 3

**Summary:**

In this paper, the authors study the power of depth for neural networks by viewing deep networks as abstract
state-transition semigroups and proving architecture-agonistic Rademacher complexity bounds for these abstract models.
This leads to a bias-variance decomposition, where the variance is the Rademacher complexity and the bias is determined
by how well actual networks can approximate those idealized abstract models.

Then, they provide several sufficient conditions under which the variance grows polynomially or logarithmically.
Combing these two cases with the exponential or polynomial decay of the bias leads to four regimes, and in each
of these regimes, the authors identify the optimal depth by equating the bias and variance terms. Finally, several
examples are given for these regimes.

**Strengths:**

* Overall, this is a clean and well-written paper.
* The idea of decoupling the implementation details and idealized networks through abstract state transitions is
  interesting and executed well.

**Weaknesses:**

The main issue of this paper is that it is more of a paper about abstract Rademacher complexity instead of deep learning
theory.

Architecture-independent Rademacher complexity bounds have gone out of fashion since around 2020, and I believe
this is for good reason. It was empirically shown in the seminal paper [1] that modern networks can fit random noises
but still generalize well if trained properly, indicating that vanilla Rademacher complexity may not be a good way
to measure generalization in deep learning. This is why most of the papers after that consider specific models
and/or algorithmic regularization, hoping to show that certain biases induced by the architecture and/or training
algorithms make the network generalizable even when the network has the capacity to fit random noises.
I do not think this paper manages to resolve/go around the above issue of implementation-agnostic generalization
bounds, as it is still based on bounding the covering number of some abstract generic deep networks.

To be fair, the authors do provide several sufficient conditions under which the covering number grows polynomially
with the depth (Sec. 4.1). However, most of them are either hard to check or impose strong constraints, and the
examples provided are also either abstract or about a very specific non-conventional problem (neural operators).

A small suggestion:
It might be better to rewrite Table 1 based on the target error. For example, suppose that the target error is $\epsilon$.
Then, we can derive the needed depth for the bias to be small and then the number of samples required to make the
variance small. The optimal generalization error is usually not the quantity of interest once it falls bellow a certain
threshold. Trying to achieve the optimal generalization error makes the depth grows as the number of samples grows,
which makes the bound look strange, since intuitively having more data should always help the model.

[1] Chiyuan Zhang, Samy Bengio, Moritz Hardt, Benjamin Recht, Oriol Vinyals. Understanding deep learning requires rethinking generalization. 2017

**Questions:**

I would be willing to raise my score if the authors can find a concrete example demonstrating the applicability of their results.
More specifically, could you find a concrete target function class (composition of quadratic networks with certain properties,
functions with a tree-like hierarchical structure, or something at a similar level of concreteness), and a class of
abstract models and their implementations, for which the results of this paper can be used to derive non-vacuous
bounds?

---

> ### Author Response · Authors · 2025-11-18
>
> We appreciate your taking time and constructive comments. We are pleased to see your positive comments on our main results.
>
> > The main issue of this paper is that it is more of a paper about abstract Rademacher complexity instead of deep learning theory.
>
> > Architecture-independent Rademacher complexity bounds have gone out of fashion since around 2020, and I believe this is for good reason. It was empirically shown in the seminal paper [1] ... I do not think this paper manages to resolve/go around the above issue of implementation-agnostic generalization bounds, as it is still based on bounding the covering number of some abstract generic deep networks.
>
> We are certainly aware of the issues raised by Zhang et al. 2017, and a series of extensive studies on regularization effects of learning process---such as double descent, flat minima, lazy learning, and benign overfitting. On the other hand, a decade later, several authors have also demonstrated that Rademacher complexity is not entirely useless---it can even yield depth-independent bounds by imposing additional assumptions on the hypothesis class $H$ or the concept class $C$. Examples include Golowich et al. 2018, 2020 and Schmidt-Hieber 2020, as reviewed in Appendix A. In other words, it's been confirmed that assuming a small hypothesis class allows the generalization error bounds to be kept low.
>
> We are rather concerned that recent deep learning theory has been confined to the analysis of linear/shallow models. While the regularization effect of the learning process described above seems plausible, the analysis is almost entirely limited to linear/shallow models. Of course, there are analyses of nonlinear deep models, but these are often confined to analyses under extremely strong assumptions, such as component-wise ReLU or perturbation approximations.
>
> One focus of this study is to precisely investigate the **effect of depth** (or function composition), and our discovery lies not only in the generality of Theorems 1+2 but also in the list of connections between growth rate and background geometric structures, as shown in Section 4. If one further wants to reflect the effect of learning process, they can simply regularize $H_k$ appropriately and run Theorem 1+2 (the current theorem holds as is).
>
> > To be fair, the authors do provide several sufficient conditions under which the covering number grows polynomially with the depth (Sec. 4.1). However, most of them are either hard to check or impose strong constraints,
>
> In the revised version, we refined the conditions in Section 4, generalizing and simplifying them to encompass the original conditions (please also refer to the general comments). In particular, the new P1 is powerful: if the domain $X$ is compact and generator $F$ is non-expansive (i.e. isometric or contractive), then the growth-rate is **depth-independent**. Therefore, to obtain non-constant growth (with depth-k), one must assume either non-compactness of $X$ or non-extensiveness of $F$ (please also refer to new Table 1 in the revised manuscript). The refined criteria are generally comprehensive and much **easier to check**.
>
> > and the examples provided are also either abstract or about a very specific non-conventional problem (neural operators).
>
> What does “conventional” mean here? For example, Holder/Sobolev/Besov classes? These traditional classes are closed under linear operations, making them suitable for analysis in the **width** direction. However, these are not closed under function composition (e.g. the composition of Holder functions is not necessarily a Holder function), making them unsuitable for analysis in the **depth** direction.
>
> In contrast, neural operators (NOs) are a prime example when the data generator (concept class $C$) has a **hierarchical structure**. This is because differential equations (vector fields) are closed under both composition and linear operations. For instance, as also reviewed in Section 6, Schmid-Hieber 2020 and subsequent work like Fukumizu-Imaizumi also highlight the importance of hierarchical structures in data generators. However, these prior studies only considered the extremely limited condition of composition of Holder functions. NOs can represent a broader range of function systems than these studies. Therefore, we consider NOs to be a highly conventional setting when considering deep learning use cases.
>
> > A small suggestion: ...
>
> We appreciate your constructive suggestion. We will update the table by target error. (Response may be delayed due to work commitments, but please rest assured it has not been ignored.)
>
> > **Questions:** ...
>
> Please review the updated conditions. The Cantor set examples (Examples 3 and 11) can be interpreted as clear illustrations that mere tree-like structure alone does not determine growth rate.

---

> > ### Comment · Reviewer_Z8EL · 2025-11-23
> >
> > I thank the authors for the response. While I personally still don't agree with the authors' abstract approach, I appreciate their effort in refining the conditions and providing more examples in the revision, and believe that it might interest a certain subset of the ICLR audience and could potentially be useful to them. Therefore, I'll raise my score to 6.
> >
> > When I said "concrete examples", the models I had in mind were the deep quadratic model in [1] and the PCFG model in [2]. In both cases, depth is essential and the models themselves are simple enough to potentially be used as a testbed for other results. I am also OK with adding more restrictions on the target class, as long as they don't make the learning task trivial.
> >
> > [1] Zeyuan Allen-Zhu, Yuanzhi Li. Backward Feature Correction: How Deep Learning Performs Deep (Hierarchical) Learning. 2023
> >
> > [2] Francesco Cagnetta, Leonardo Petrini, Umberto M. Tomasini, Alessandro Favero, Matthieu Wyart. How Deep Neural Networks Learn Compositional Data: The Random Hierarchy Model. 2023

---

> > > ### Author Response · Authors · 2025-12-01
> > >
> > > Thank you for your positive comments.
> > >
> > > And thank you as well for providing concrete examples---they are extremely interesting directions. I had previously thought that polynomial functions and random weights could be regarded merely as perturbative approximations of the initial state (rather than the final state) and thus were not worth investigating deeply, but your comments have sparked my interest in exploring them further.

---

### Official Review · Reviewer_j2jb · 2025-10-27

**Soundness:** 3
**Presentation:** 1
**Contribution:** 2
**Rating:** 4
**Confidence:** 2

**Summary:**

This paper develops a theoretical framework for understanding when and why deeper neural networks outperform shallow ones. The key innovation is formulating deep networks as abstract state-transition systems on metric spaces, separating the mathematical analysis from specific implementations. The authors prove a bias-variance decomposition where variance depends only on the abstract depth-k network structure, then analyze how covering numbers of "word balls" (compositions of k functions) grow with depth. They identify four regimes (EL/EP/PL/PP) based on whether bias decays exponentially or polynomially and whether variance grows logarithmically or polynomially with depth. The framework provides explicit optimal depths and explains why depth helps most for hierarchical/iterative tasks like neural ODEs, diffusion models, and chain-of-thought reasoning.

**Strengths:**

- The approach taken in the paper appears to be genuinely novel and of interest to mathematicians.
- The mathematics introduced in the paper is backed up by extensive proofs and discussion.
- The task of identifying optimal depths of neural networks is of great importance to the field.

**Weaknesses:**

- The relationship between theoretical concepts to real neural networks is weak
- There are no genuine examples provided to assist readers in understanding the practical benefits of the work
- The paper is very mathematically dense, reducing its overall impact on a broad community. This work feels more at home in a mathematics journal than a broader conference. It gives the impression that little to no effort was made to make the work accessible to the majority of machine learning practitioners.

**Questions:**

- In the paper, the authors write, "In practice, trained networks obtained by optimization are not
identical to exact empirical risk minimizers, but we assume they are identical for simplicity". This appears to be quite a substantial approximation. Can the authors shed light on how much of a difference one can expect in a real setting where gradient descent is used to train models?
- Were any practical experiments performed, perhaps on small networks, to validate the theory shown here in practice?
- Given that the authors claim to have found a means of selecting optimal depths of networks, how does one practically apply this to their models today?

---

> ### Author Response · Authors · 2025-11-18
>
> We appreciate your taking time and feedback. We are pleased to see your positive comments on our main theorems and subsequent discussions.
>
> > The relationship between theoretical concepts to real neural networks is weak
>
> Theorem 1 incorporates approximation errors arising from implementation, enabling it to be combined with specific network approximation error analysis. Furthermore, Section 6 discusses diverse concrete examples.
>
> > There are no genuine examples provided to assist readers in understanding the practical benefits of the work
>
> In the revised version, we refined the conditions in Section 4, generalizing and simplifying them to encompass the original conditions (please also refer to the general comments). In particular, the new P1 is powerful: if the domain $X$ is compact and generator $F$ is non-expansive (i.e. isometric or contractive), then the growth-rate is **depth-independent**. Therefore, to obtain non-constant growth (with depth-k), one must assume either non-compactness of $X$ or non-extensiveness of $F$ (please also refer to New Table 1 in the revised manuscript). The refined criteria are generally comprehensive and much **easier to check**.
>
> > The paper is very mathematically dense, reducing its overall impact on a broad community. This work feels more at home in a mathematics journal than a broader conference. It gives the impression that little to no effort was made to make the work accessible to the majority of machine learning practitioners.
>
> Thank you for your feedback. The ICLR CfP lists *learning theory* as one of their Subject Areas, and this is a deep learning theoretical study, so we selected learning theory as the Primary Area at submission. Deep learning theory has accumulated over a decade of research (as summarized in Appendix A), making it inherently dense, but we strive to make it useful for practitioners as well. As also acknowledged by other reviewers, Theorems 1+2 and the subsequent sections contain numerous technically novel and useful results.
>
> > **Questions:**
> > In the paper, the authors write, "In practice, trained networks obtained by optimization are not identical to exact empirical risk minimizers, but we assume they are identical for simplicity". This appears to be quite a substantial approximation. Can the authors shed light on how much of a difference one can expect in a real setting where gradient descent is used to train models?
>
> The focus of this study is on the detailed analysis of **estimation errors**, so we have not pursued specific approximation/optimization errors. Evaluating optimization errors, in particular, requires analysis specific to the network implementation, which constitutes research substantial enough for another single paper and is therefore out of scope. For example, if one wishes to analyze cases where optimization converges near the initial parameters, one can assume a small space $H_k$ consisting only of models near the initial parameters. This allows Theorems 1+2 and the subsequent analysis to be applied.
>
> > Were any practical experiments performed, perhaps on small networks, to validate the theory shown here in practice?
>
> We did not conduct any experiments.
>
> > Given that the authors claim to have found a means of selecting optimal depths of networks, how does one practically apply this to their models today?
>
> Examples are provided in Section 5.

---

> > ### Comment · Reviewer_j2jb · 2025-11-24
> > **Response to authors by Reviewer j2jb**
> >
> > I appreciate the responses from the authors. The responses were thoughtful and addressed some of the concerns raised in the initial review. The only additional comment I would like to stress is related to the defence of the mathematical density of the work. Learning theory is a very dense topic, much like many other mathematical fields, however, that doesn't mean that it can't be explained in a way that makes it approachable by a broad audience. If the authors don't want that reach, that is okay, but it doesn't mean it is impossible. Based on the response and given the confidence level expressed in the initial review, there is no clear reason to change the initial score.

---

> ### Author Response · Authors · 2025-11-26
>
> > given the confidence level expressed in the initial review,
>
> We are sorry, but we couldn't quite understand the English. Whose confidence level is this? We have never expressed a confidence level.
>
> > broad audience
>
> This is a highly subjective opinion and should be avoided in peer reviews whenever possible.

---

### Official Review · Reviewer_pQhN · 2025-10-29

**Soundness:** 3
**Presentation:** 3
**Contribution:** 3
**Rating:** 4
**Confidence:** 3

**Summary:**

This paper provides an implementation-agnostic explanation of why and when deep networks outperform shallow ones through a novel bias–variance decomposition. The authors show that while the variance term depends only on the depth-k state-transition structure (not on the network type), the bias decays with depth—often exponentially—leading to optimal finite depths $
k* >1$. They identify four canonical bias–variance regimes and show that exponential bias decay with logarithmic variance growth (the EL regime) explains depth supremacy in hierarchical or iterative models such as neural ODEs, diffusion models, and chain-of-thought reasoning.

**Strengths:**

The paper is well written and easy to follow. Analyzing the generalization error of neural networks through a bias–variance decomposition is a compelling and insightful approach, offering a clear framework to understand the benefits of depth.

**Weaknesses:**

Although the contribution is interesting, several aspects remain unclear to me. Please see my questions below.

**Questions:**

-**Section 2.3**: I am confused about the role of the embedding operator — does it address optimization difficulties or something else? The authors state ``In practice, trained networks obtained by optimization are not  identical to exact empirical risk minimizers, but we assume they are identical for simplicity.'' It is not clear to me why this section is titled “Regularized Empirical Risk Minimization”; the connection to regularization is not well explained, and the section feels conceptually confusing.
I would suggest that authors explain this section and the different sources of errors clearly.

-**Section 3.3**: I am confused about the novelty of Theorem 1. How does this result differ from or extend the existing results presented in Mohri’s book? Foundations of Machine Learning, Chapter 3.

-**Theorem 1**:  I think the error term  $ε_{imp}$ should also depend on the structure of the network space. Please clarify.

-While the authors analyze the variance across different structural settings, it is not clear how the approximation (bias) error behaves exactly. Or did I perhaps overlook that part?

---

> ### Author Response · Authors · 2025-11-18
>
> Thank you for your careful reading. We are pleased to see your positive comments on our results.
>
> > - **Section 2.3:** I am confused about the role of the embedding operator — does it address optimization difficulties or something else?
>
> The embedding operator is independent of optimization difficulty. This is imposed for rather a technical assumption to separate the effects of specific parameterizations and depth $k$ in the analysis of **estimation error** (or the **variance** term  $R_S(H_k)$).
>
> > It is not clear to me why this section is titled “Regularized Empirical Risk Minimization”
>
> Because we consider a two-stage learning process: first, performing loss minimization on an ideal class $H_k$, which is not assumed to have specific parameterizations, then projecting the minimizer onto an implementation class $H_{imp}$, which is assumed to have specific parameterizations. The first step is restricted to $H_k$ (rather than entire $H_{imp}$), which is regarded as regularization, hence the term “Regularized” (regularization generally refers to operations that suppress the complexity of the hypothesis space).
>
> We consider this is not a strong assumption because typical neural networks have the universal approximation property. So the error associated with embedding (from $H_k$ to $H_{imp}$) is negligible in typical settings, and thus the two-stage learning result is a good proxy to the solution obtained by optimizing on $H_{imp}$ (by further assuming that ).
>
> However, to avoid unnecessary confusion, we will simply refer to this as ``two-stage learning'' in the revised manuscript.
>
> > - **Section 3.3:** I am confused about the novelty of Theorem 1. How does this result differ from or extend the existing results presented in Mohri’s book? Foundations of Machine Learning, Chapter 3.
>
> In this study, we set up **two** hypothesis classes, $H_{imp}$ and $H_k$, and perform two-stage learning. Mohri's Theorem 3.3 (more closely related is Theorem 4.2 for ERM) does not assume such a two-stage structure. Therefore, the estimation error term becomes $R_S(H_{imp})$---the Rademacher complexity of **implementation class**---where the implementation effects explicitly appear in the estimation error, complicating the analysis. In Theorem 1, through careful computation, we show that the estimation error term becomes $R_S(H_k)$---Rademacher complexity of **abstract class**. That is, we successfully eliminate the effects originating from the implementation. As the proof of Theorem 1 demonstrates, this is not trivial, and it is indeed a new theorem.
>
> > - **Theorem 1:** I think the error term should also depend on the structure of the network space. Please clarify.
> > - While the authors analyze the variance across different structural settings, it is not clear how the approximation (bias) error behaves exactly. Or did I perhaps overlook that part?
>
> As you pointed out, the approximation error depends on the network implementation. This is (not carelessly omitted but) **reflected** in the implementation error $\varepsilon_{imp}$. Since the focus of this study is on the detailed analysis of the estimation error, we do not further investigate specific approximation error but simply assume that a sufficiently small approximation error can be achieved.
>
> Indeed, in many typical cases, neural networks possess the universal approximation property, so $\varepsilon_{imp}$ can be made sufficiently small. For example, if $X$ is a $d$-dimensional compact manifold (a highly realistic setting), it can be (smoothly) embedded as a compact subset of an $m (> 2d)$-dimensional Euclidean space by Whitney embedding theorem. Thus, the classical universal approximation property of a fully connected neural network with single hidden layer network over $\mathbb{R}^m$ allows achieving arbitrarily small embedding error. One point to note is that while the width of such a network may be very large, Theorem 1 shows that the estimation error is independent of the implementation, and thus does not affect the estimation error. This separation of estimation error from implementation dependency is the core of Theorem 1.

---

> > ### Comment · Reviewer_pQhN · 2025-11-26
> >
> > Thanks to the authors for their response.
> > Regarding Theorem 1, I believe this is a straightforward observation: the error is decomposed into the approximation error
> > $\epsilon_{emp}$ (which is then ignored by the approximation power of NN), and a second error term depending on the complexity of $H_k$ (following Eq 9--13 in your proofs), which can follow directly from Mohri’s result.
> > I invite the authors to correct me if I am wrong, as Theorem 1 is introduced as the main result of the paper.
> > For this reason, I would keep my score unchanged for now.

---

> > > ### Author Response · Authors · 2025-11-26
> > >
> > > Thank you for your comments. Theorem 1 is a first step of our contribution, but it is not the entirety of what this study contributes. We believe that whether Theorem 1 itself is novel or original is not the essential question in assessing our contribution.
> > >
> > > For completeness, we recall Theorem 3.3 from Mohri’s textbook below:
> > >
> > > Let $G$ be a family of functions mapping from $Z$ to $[0,1]$. Then, for any $\delta>0$, with probability at least $1−\delta$ over the draw of an i.i.d. sample $S$ of size $m$, each of the following holds for all $g \in G$:
> > > \\[
> > > \begin{aligned}
> > > E[g] &\le \frac{1}{m} \sum_{i=1}^m g(z_i) + 2 R_m(G) + \sqrt{\frac{1/\delta}{2m}} \\\\
> > > E[g] &\le \frac{1}{m} \sum_{i=1}^m g(z_i) + 2 \hat{R}_S(G) + 3 \sqrt{\frac{1/\delta}{2m}}
> > > \end{aligned}\\\]
> > >
> > > This is a version of bias-variance decomposition that was known before Mohri, and Mohri’s textbook is not the original source of this result.
> > >
> > > To reiterate, Theorem 1 is a foundational result used to develop the subsequent theory; it is not, by itself, the main claim of this work. The subsequent Theorem 2 and Sections 4, 5, and 6 contain content that is independent of Mohri’s Theorem 3.3. We would therefore kindly ask that the paper be evaluated based on the entirety of its contributions.

---

### Official Review · Reviewer_vYAF · 2025-11-01

**Soundness:** 4
**Presentation:** 3
**Contribution:** 2
**Rating:** 4
**Confidence:** 3

**Summary:**

This paper provides a framework to prove bounds on the generalization performance of deep neural networks.

The setting is abstracted as learning a function in a function class which consists of a composition of blocks lying in some $F \subset C(X,X)$, composed with a read-out head in $H \subset C(X)$.

A novel bound combining Rademacher complexity of H, plus a Dudley-style integral involving the covering numbers of $B_k := \underbrace{F \circ F \circ \dots F}_{k \mbox{ times }}$, is proved.

Subsequently, the paper presents several sufficient conditions for the covering numbers of $B_k$ to be either $O(1)$ or $\poly(k)$ or $\exp(O(k))$ as $k$ grows.

Finally, the paper considers the tradeoff between the bias term (deeper architectures yield lower bias because they are more expressive), and the variance term (deeper architectures yield higher variance because they are more expressive), and derives an optimal depth based on this tradeoff.

**Strengths:**

The theoretical results are precisely presented and appear to be sound to the best of my knowledge.

Theorem 2 provides a nice way to conveniently bound the generalization error.

In Section 4, there is a plethora of sufficient conditions under which the covering number of $B_k$ remains bounded. These involve nice connections with various other branches of mathematics.

**Weaknesses:**

* Overall, I do not see what is fundamentally different between this work and prior works, beyond proposing a more general framework. The introduction's motivation of the paper is that classical measures of generalization suggest that generalization should worsen with depth, and this paper provides an alternative perspective. However, I am not convinced that this paper provides a truly different view.

    - The bounds on covering number of the class in this paper increase with depth (similarly to generalization error bounds growing with depth in the prior work that this paper cites).
    - This paper gets around this by positing that the bias decreases with depth, and computing the depth for an optimal a bias-variance tradeoff. But the idea that bias might decrease with depth is the motivation of past works that prove expressivity separations between neural networks of different depths.

* I am also not convinced that providing the generalization bounds in this work (based on covering number) are the most promising path to understand why networks generalize. Many networks in practice, such as image networks are overparametrized, and can memorize random labels when trained. See, e.g. "Understanding deep learning requires rethinking generalization" by Zhang et al., 2016. Therefore, the optimization must be providing some regularization that allows the network to learn a simple solution. Getting a concrete understanding of what that regularization is seems to be a more promising path.

**Questions:**

See weaknesses:
* How is the approach fundamentally different from prior works providing generalization bounds?

---

> ### Author Response · Authors · 2025-11-18
>
> We appreciate your taking time and productive comments. We are pleased to see your positive comments on Theorem 2 and Section 4.
>
> > **Questions:**
> > How is the approach fundamentally different from prior works providing generalization bounds?
> >
> > **Weaknesses:**
> > - The bounds on covering number of the class in this paper increase with depth (similarly to generalization error bounds growing with depth in the prior work that this paper cites).
> > - This paper gets around this by positing that the bias decreases with depth, and computing the depth for an optimal a bias-variance tradeoff. But the idea that bias might decrease with depth is the motivation of past works that prove expressivity separations between neural networks of different depths.
>
> The contribution of this study is certainly not rediscovering the bias-variance tradeoff.
> The monotone-increase of covering numbers is of course trivial by definition (note that $B_k$ is defined as the union $B_k = \cup_{i=0}^k F^i$ rather than a single $F^k$, as you described in your summary, so it must increase).
> - Theorems 1+2 separate the effects of depth and specific parametrizations in generalization error analysis, which is technically useful and constitutes a solid contribution to machine learning theory.
> - Theorems 1+2 mathematically clarify that generalization can be recast as coarse geometry and geometric group theory, where the growth-rate of a metric space is a central concept.
> - Section 4 investigates in detail the rate at which the covering number grows, revealing the simple geometric structure underlying previous case-by-case studies.
>
> > - I am also not convinced that providing the generalization bounds in this work (based on covering number) are the most promising path to understand why networks generalize. ... See, e.g. "Understanding deep learning requires rethinking generalization" by Zhang et al., 2016. Therefore, the optimization must be providing some regularization that allows the network to learn a simple solution. Getting a concrete understanding of what that regularization is seems to be a more promising path.
>
> We are well aware of the issues raised by Zhang et al. 2016, and a series of extensive studies on regularization effects of learning process---such as double descent, flat minima, lazy learning, and benign overfitting. On the other hand, a decade later, several authors have also demonstrated that Rademacher complexity is not entirely useless---it can even yield depth-independent bounds by imposing additional assumptions on the hypothesis class $H$ or the concept class $C$. Examples include Golowich et al. 2018, 2020 and Schmidt-Hieber 2020, as reviewed in Appendix A. In other words, it's been confirmed that assuming a small hypothesis class allows the generalization error bounds to be kept low.
>
> We are rather concerned that recent deep learning theory has been confined to the analysis of linear/shallow models. While the regularization effect of the learning process described above seems plausible, the analysis is almost entirely limited to linear/shallow models. Of course, there are analyses of nonlinear deep models, but these are often confined to analyses under extremely strong assumptions, such as component-wise ReLU or perturbation approximations.
>
> One focus of this study is to precisely investigate the **effect of depth** (or function composition), and our discovery lies not only in the generality of Theorems 1+2 but also in the list of connections between growth rate and background geometric structures, as shown in Section 4. If one further wants to reflect the effect of learning process, they can simply regularize $H_k$ appropriately and run Theorem 1+2 (the current theorem holds as is).
>
> We would be delighted if you could reassess the strength of this study.

---

### Author Response · Authors · 2025-11-18
**We appreciate feedbacks. Revised the manuscript.**

We appreciate the constructive comments from all the four reviewers. We are pleased that the novelty, functionality, and generality of our Main Theorems 1+2 have been recognized.

We have **revised the manuscript**,  especially **Section 4**. We refined the sufficient conditions listed in Section 4, organizing them into **simpler and more comprehensive** conditions. Below, we outline the correspondence between the submitted version and the revised version (please also refer to the New Table 1 in the revised manuscript).

| revised | original | comment |
| :--- | :--- | :--- |
| P1 + P1' | P1, P2, P2', P4, P5 | depth-independent rates are unified |
| P2 | P3, P3' | re-numbered (from P3 to P2) |
| E1 | E2 | generalized and re-numbered (from E2 to E1) |
| E2 | E1 | generalized and re-numbered (from E1 to E2) |
| E3 | - | super-exponential rates are newly supplemented |
| - | E3 | original E3 was deleted as it was a bit trivial |

In particular, **new P1** is a powerful condition: if the domain $X$ is compact and generator $F$ is non-expansive (i.e. isometric or contractive), then a depth-independent rate is concluded because the infinite-depth ball $\langle F \rangle := B(\infty,F)$ converges to a  compact set, as a consequence of the Arzela-Ascoli theorem.
Therefore, to obtain non-constant growth (with depth-k), one must assume either non-compactness of $X$ or non-extensiveness of $F$ (see New Table 1 in revised manuscript). A striking consequence of this result is that even a typically **free generators like the Cantor map** yields a **depth-independent rate**, as shown in a simple argument (Example 3). Furthermore, it is demonstrated that the same Cantor map exhibits **exponential growth** when it is expansive (Example 11).

---

> ### Author Response · Authors · 2025-12-01
> **Summary for our new AC**
>
> Dear New AC,
>
> Thank you very much for taking on this role. Let us briefly summarize the situation so far.
>
> The initial scores were 4-4-4-4. During the discussion period, some scores were raised.
>
> This study proposes a new direction for estimating generalization error in deep learning.
> Conventionally, generalization error analysis depends on the particular implementation of the network, so when the network implementation changes, a new analysis is required.
> Our method carefully factors out implementation-specific effects and instead investigates in detail the relationship between the geometry of the data space $X$ and hidden layers $F$ and the generalization error bound.
>
> The development of deep learning theory has already spanned more than ten years, and many different approaches have been explored.
> Our method is a **Rademacher-based** analysis, and our goal is to fully exploit its power through geometric and semigroup-theoretic analysis.
> Regarding the fact that our method is Rademacher-based, two reviewers (**vYAF** and **Z8EL**) raised the following concern:
> they argued that it does not align with the critique by Zhang et al., namely that Rademacher complexity does not reflect the learning process and is therefore inappropriate for bounding the generalization error of deep networks.
>
> However, this is a **misunderstanding**. As shown by the work following Zhang (summarized in **Appendix A** and **Section 6**), by computing the Rademacher complexity restricted to the set of solutions reached by the learning process, Rademacher complexity can in fact reflect the effect of the learning dynamics. Our proposed method also reflects the effect of the learning process in this way.
> On this point, reviewer **Z8EL** expressed understanding, and their score was improved.
> In the case of reviewer **vYAF**, the system failure closed the discussion before we could receive their post-rebuttal comments, but given that they assigned a 4 for Soundness, we believe their assessment is positive and we expect that they would have raised their score as well.
>
> As explained in the General Comments, in the revised manuscript we refined the conditions in **Section 4** (the part corresponding to the examples of Main Theorems 1 and 2) and increased their **mathematical coverage**.
> In particular, the **new P1** is, to our knowledge, the first to identify a simple yet comprehensive condition for obtaining “depth-independent” bounds, which represents a significant contribution to deep learning theory. This contribution was recognized by **Z8EL**, and their score was improved.
> Had the discussion period continued, we expect that the other reviewers would also have expressed positive views about this contribution.
>
> From reviewers **pQhN** and **j2jb**, there were no concrete comments on the main contributions of the paper (Theorems 1 and 2, Sections 4–6) at the initial review stage. We were just about to begin a more in-depth discussion during the discussion period, but before that could happen, the responses were closed due to the system failure.
>
> In light of the above, we would be grateful for your overall assessment of the paper.

---

### Meta-Review · Area_Chair_QZsA · 2025-12-28

**Summary:**

This paper considers an abstract approach to understand the optimal depth for neural networks. The approach considers an abstract state-transition semigroup and gives new variance bounds through Rademacher complexity and covering numbers. Using the idea of bias-variance tradeoff the paper can compute optimal depth k^* for several regimes and k^* is often more than 1. The reviewers raised several concerns. The main concern is that the covering-number based bounds are just upperbounds and it's not clear how such a bound reflects the true generalization error. This is partially addressed by the authors by pointing out relevant results showing such bounds can be reasonable under certain assumptions.

**Reviewer Concerns:**

The main concern is that the covering-number based bounds are just upperbounds and it's not clear how such a bound reflects the true generalization error. Author provided some justification and reviewer Z8EL seems convinced. On the other hand, the existence of cases where Rademacher complexity gives depth-independent bounds doesn't necessarily imply any time a network achieves small generalization error the bound on Rademacher complexity would be good. The bounds are still just upperbounds. The authors mentioned many works for recent learning theory focuses on shallow and linear cases, which is a fair criticism, but the effects of these paper has been empirically observed for deeper, nonlinear networks. Just because we don't know how to do the proof in those cases doesn't mean those effects don't happen.

**Reviewer Scores:**

vYAF: may or may not change as the authors justification doesn't completely address the concern.
pQhN: most concerns partially addressed, may change to 5
j2jb: has a very different mindset, may not change the score.
Z8EL: score changed to 6

---

### Decision · Program_Chairs · 2026-01-26

Reject